## nature
## ecology & evolution
## OPEN

# Widespread extinction debts and colonization credits in United States breeding bird communities

Yacob Haddou [1], Rebecca Mancy [1,2], Jason Matthiopoulos [1], Sofie Spatharis [1,3] and Davide M. Dominoni [1] ✉

**Species extinctions and colonizations in response to land cover change often occur with time lags rather than instantaneously, leading to extinction debts and colonization credits. These debts and credits can lead to erroneous predictions of future biodiversity. Recent attempts to measure debts and credits have been limited to small geographical areas and have not considered multiple land cover types, or the directionality of land cover change. Here we quantify the relative contribution of past and current landscapes on the current effective number of species of 2,880 US bird communities, explicitly measuring the response of biodiversity to increases and decreases in five land cover types. We find that the current effective number of species is still largely explained by the past landscape composition (legacy effect), depending on the type, magnitude and directionality of recent land cover change. This legacy effect leads to widespread extinction debts and colonization credits. Specifically, we reveal debts across 52% of the United States, particularly in recently urbanized areas, and colonization credits in the remaining 48%, which are primarily associated with grassland decrease. We conclude that biodiversity policy targets risk becoming rapidly obsolete unless past landscapes are considered and debts and credits accounted for.**

Preventing an irreversible loss of biodiversity is one of humanity's greatest contemporary challenges[1]. Anthropogenic habitat loss is recognized as a major driving force of species extinctions, threatening up to 85% of all species included in the International Union for Conservation of Nature Red List[2]. Current understanding of the impacts of habitat change on biodiversity is heavily reliant on the assumption that species respond rapidly to disturbances. However, the role that legacy effects and lags play in species' responses to environmental change is increasingly recognized[3–8].

The diversity of species at a given location is only partially determined by the current state of a habitat. Rather, it is the consequence of a legacy of complex historical effects of landscape change on community composition[9,10]. Notably, species' responses to changes in land cover composition are rarely instantaneous, but instead are subject to lags leading to gradual species extinctions and colonizations at the landscape scale[7,11]. Thus, current observations of biodiversity could be substantially higher than a recently modified landscape is actually able to support, generating so-called extinction debts. In the opposite scenario, recent modifications that will, in time, be favourable to biodiversity, could instead lead to colonization credits.

Extinction debts and colonization credits form the focus of a growing area of research in community ecology[7,11], but have rarely been explicitly incorporated into predictive models of biodiversity over large spatial scales[4,5,12]. This hinders our ability to correctly quantify future biodiversity loss and increases the risk of policy strategies becoming out-of-date before they are even introduced[8]. Moreover, the type and directionality of habitat change may result in different magnitudes of legacy effects and lagged responses, leading to spatial variation in debts and credits. However, so far, most studies have focused on the loss of either forests or grasslands, largely ignoring gains and other habitat types[3,4,12,13]. To generate predictions of biodiversity that can reliably inform environmental policies, the contribution of different types of past landscapes and subsequent legacy effects on the composition of current communities need to be quantified and incorporated into large-scale spatio-temporal models. Here we developed such a model using bird diversity data collected from 2,880 bird communities over a 15-year period in the contiguous USA and validated our predictions using independent data from a more recent survey.

## Results

**Modelling extinction debts and colonization credits.** Birds are an ideal taxon for analyses of spatial and temporal biodiversity changes because they have long been monitored over broad spatial scales and are highly sensitive to anthropogenic disturbance[15]. We calculated the species diversity of 2,880 communities surveyed as part of the North American Breeding Bird Survey (BBS, Extended Data Fig. 1), which comprises information on the abundance of more than 500 bird species across the contiguous USA[14]. We defined a community as the assemblage of birds associated with the landscape surrounding each survey unit (Extended Data Fig. 2) (that is, not a prespecified habitat type). We chose the effective number of species rather than species richness as a diversity metric because it provides a more robust measure that is less sensitive to species rarity and detectability than species richness[16,17]. We also sourced high spatial resolution (30 m²) land cover data from the National Land Cover Database (NLCD)[18], as well as temperature data (mean across May and June) from the PRISM climate dataset[19] (Supplementary Figs. 1, 2 and Table 1). Using these datasets, we developed and fitted a Bayesian generalized mixed effects model (GLMM) describing the effective number of species in 2016 as a function of the weighted contribution of the landscape composition in 2016 and the past landscape composition in 2001 (hereafter, legacy model, as it incorporates information about both present and past landscape compositions). We then fitted a similar model only considering the landscape composition of 2016 (hereafter, equilibrium model, as it models the biodiversity we would expect in an equilibrium state of a

[1]Institute of Biodiversity, Animal Health and Comparative Medicine, University of Glasgow, Glasgow, UK. [2]Social and Public Health Sciences Unit, University of Glasgow, Glasgow, UK. [3]School of Life Sciences, University of Glasgow, Glasgow, UK. ✉e-mail: davide.dominoni@glasgow.ac.uk

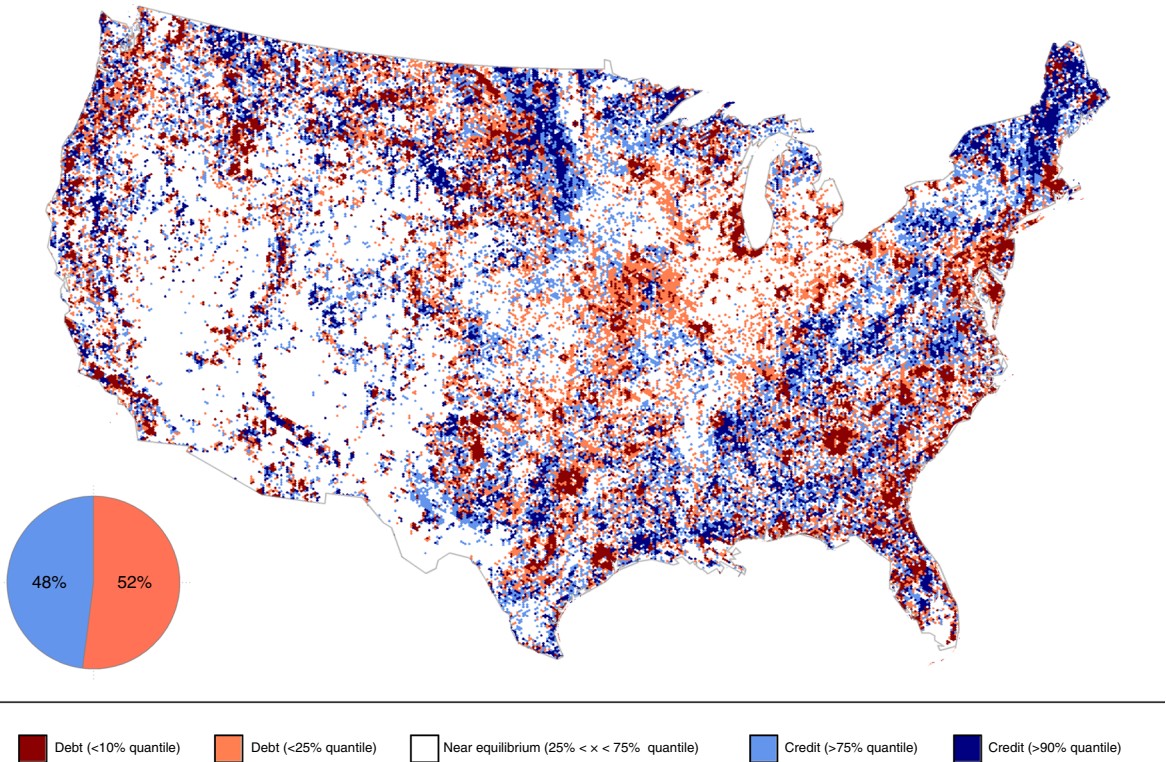

**Fig. 1 | Extinction debts and colonization credits across US bird communities.** The estimated distribution and magnitude of extinction debts (red) and colonization credits (blue) across the contiguous USA. Debts and credits were calculated by subtracting the effective number of species predicted by the legacy model from that predicted by the equilibrium model. We estimated that 48% of the contiguous USA land area is, as of 2016, experiencing colonization credits (equilibrium model − legacy model > 0), whereas 52% of it is experiencing extinction debts (equilibrium model – legacy model < 0). Note that the percentages shown in the pie chart are not the same as the map legend, which instead shows the 10% and 25% quantiles for both credits and debts. Uncertainty associated with these predictions is presented in Extended Data Fig. 4.

Legend: Debt (<10% quantile) · Debt (<25% quantile) · Near equilibrium (25% < × < 75% quantile) · Credit (>75% quantile) · Credit (>90% quantile)

static landscape with no legacies). By subtracting the effective number of species predicted by the legacy model from that predicted by the equilibrium model, we determined that colonization credits were present if the difference was positive, and vice versa, extinction debts were present if the difference was negative. A lower effective number of species in the equilibrium model highlighted an extinction debt, whereas a lower number in the legacy model spotlighted a colonization credit.

Our fitted legacy model was able to accurately predict the observed effective number of species in 2016 (Pearson correlation test, $r = 0.65$, d.f. $= 4,798$, $P < 0.01$; Supplementary Fig. 3). We further validated the model with more recent bird data from 2019[14], to confirm that the predicted debts and credits matched recently observed changes in effective number of species from 2016 to 2019. Without using any land cover change information from the same period, and despite the relatively short time interval (we expect that most of these debts and credits will require more than 3 years before they become fully realized), changes in the effective number of species since 2016 have overall been in the direction predicted by our model (Pearson correlation test, $R = 0.28$, d.f. $= 4,233$, $P < 0.001$; Extended Data Fig. 3).

Our analysis revealed the previously unknown extent of debts and credits across large areas of the contiguous USA (Fig. 1). Overall, 52% of this area is expected to lose species (extinction debts) and 48% to gain species (colonization credits) (Fig. 1). Strikingly, many of the predicted debts are localized around metropolitan areas (for example, Atlanta, Orlando, Chicago, Indianapolis, St. Louis and Houston). Conversely, predicted colonization credits are largely concentrated in the Northeast, along the Appalachian Mountains and in less inhabited areas across the country. Neglecting such debts

and credits could lead, in some locations, to overestimates of the effective number of species that a landscape can support by up to 42%, whereas in other locations, to underestimates of up to 62%.

**The past landscape casts a shadow on current biodiversity.** The debts and credits identified by our analysis originate from the substantial contribution of the past landscape to the current effective number of species (Fig. 2). Together, our results indicate that legacy effects are strong and pervasive for all land cover types, even for small magnitudes of change during a 15-year window. Indeed, a mere 10 % increase or decrease in any land cover type leads to a substantial weighting of the past land cover composition in explaining the current effective number of species (proportional contribution of past landscape >0.6; Fig. 2f). Specifically, strong legacies were observed for gains of urban and cropland (Fig. 2a,e), and for losses of grassland and cropland (Fig. 2d,e). A 10% change in these land cover types led to the effective number of species being almost completely explained by the past land cover composition (proportional contribution of past landscape ≥0.9). Conversely, a loss of 10% of forest cover or a 10% gain of grassland were associated with less pronounced legacy effects (proportional contribution of past landscape = 0.68 and 0.67, respectively; Fig. 2b,d). Whether cover was lost or gained also mattered for legacy effects; for example, forest cover gain implied a stronger legacy effect than forest loss (Fig. 2b), while the opposite was true for grasslands. Taken together, our results highlight the importance of considering multiple attributes of land cover change over time: magnitude, type and directionality.

**Explaining spatial variation in debts and credits.** Land cover changes have not been homogeneous across the contiguous USA

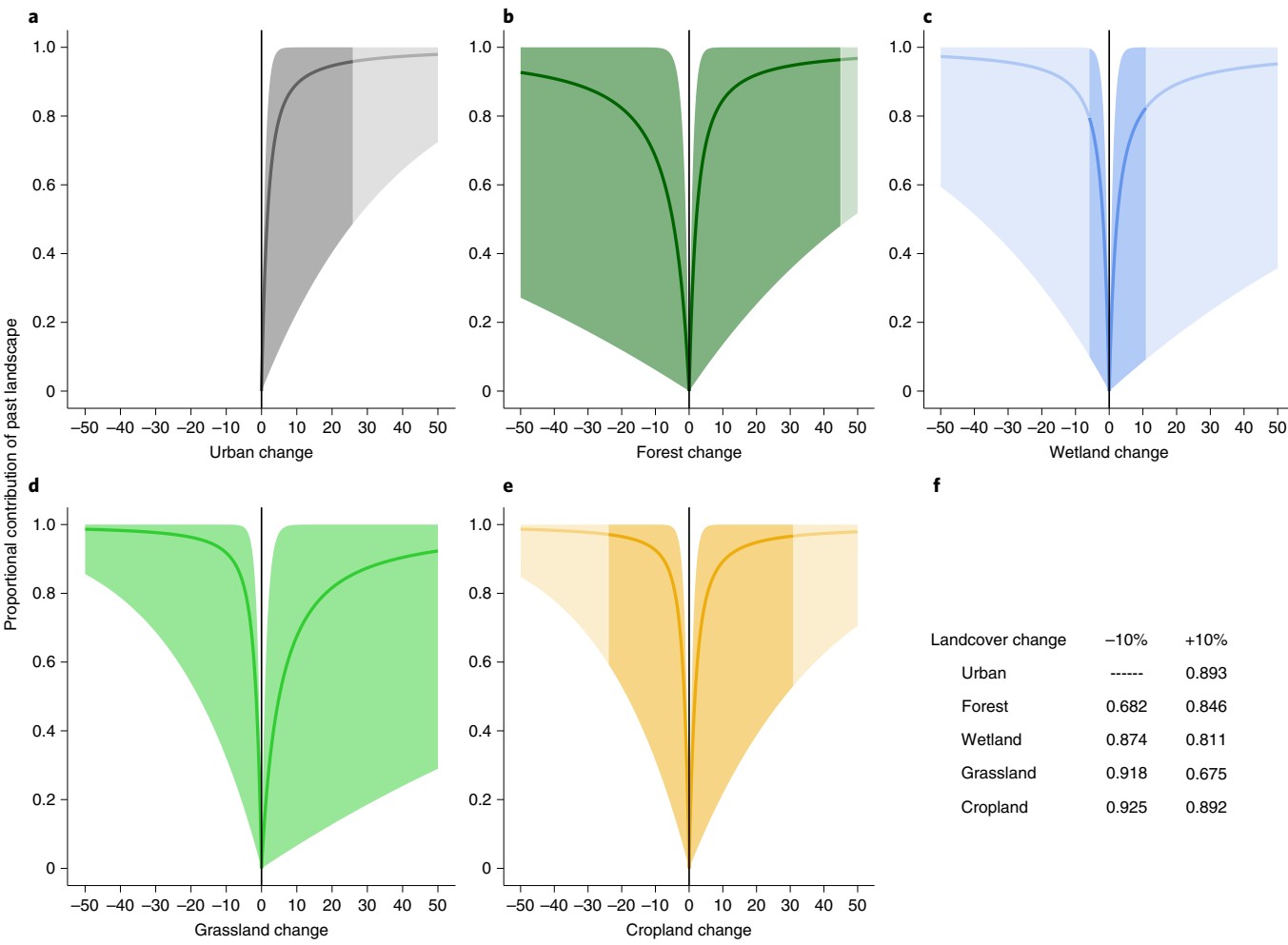

**Fig. 2 | The contribution of past landscape to the current effective number of species depends on the type, amount and directionality of land cover change. a–e,** Our model allowed us to quantify the proportional contribution of the past landscape in 2001 on the effective number of species in 2016 (y axis), in response to positive and negative changes of urban (**a**), forest (**b**), wetland (**c**), grassland (**d**) and cropland (**e**) land cover types between the two timepoints (x axis). A value of 0 on the y axis indicates that the effective number of species in 2016 is completely explained by contemporary land cover, whereas a value of 1 indicates that it is fully described by the land cover in 2001. All values presented are predictions under the assumption that no other land cover changes take place. Lines indicate the estimated mean value of the contribution of the past landscape, while coloured areas around each line represent 95% credible intervals. Lighter shaded regions are predictions outside of the maximum observed land cover change. **f,** Values of the proportional contribution of past landscape associated with a 10% increase or decrease for each land cover analysed.

(Fig. 3). For instance, much of the area in the central US has experienced large-scale conversion of grasslands (Fig. 3d) into croplands (Fig. 3e). Forest loss has been concentrated in the Northwest, as well as along the Appalachian Mountains (Fig. 3b) where forests have been mostly converted to grasslands (Fig. 3d), including pasture. Urban development has occurred around major metropolitan areas across the entire US, although particularly in the East (Fig. 3a). In contrast, the vast areas of desert and shrubland of the Southwest have experienced only very limited land cover change. We hypothesized that some of the spatial patterns in debts and credits predicted by our model should reflect this spatial segregation of different types of land cover changes. To test this hypothesis, we modelled the magnitude of the predicted debts and credits as a function of changes in land cover. We found extinction debts to be significantly associated with urban and cropland gain, and with loss of wetland (Fig. 4 and Supplementary Table 3). This is consistent with earlier findings that increases in cropland and urban cover are associated with declines in bird diversity[20,21]; similarly, wetlands are important habitats for birds[22] and it is thus unsurprising that recent wetland loss is associated with extinction debts. We found that colonization credits were

only significantly associated with recent loss of grasslands. While this association might appear at first surprising, it could be a consequence of the inclusion of pasture within the grassland category of NLCD: because pastures are globally associated with reduced animal diversity[23], the reduction of grassland might result in future benefits to biodiversity.

## Discussion

By quantifying the geographical extent and magnitude of debts and credits, we have revealed the invisible footprint of anthropogenic change on bird biodiversity at continental scale. Far from being a minor effect, we estimate that the contiguous USA is already committed to biodiversity changes, of different magnitudes, that have yet to become realized. Moreover, we emphasize that the legacy of past landscapes on the current biodiversity (effective number of species) is dependent not only on the type and amount of land cover change, but also on its directionality. By accounting for all these aspects in our model, we show the expected widespread distribution of future species extinctions and colonizations across a large geographical area. Our results spotlight areas of conservation concern, particularly

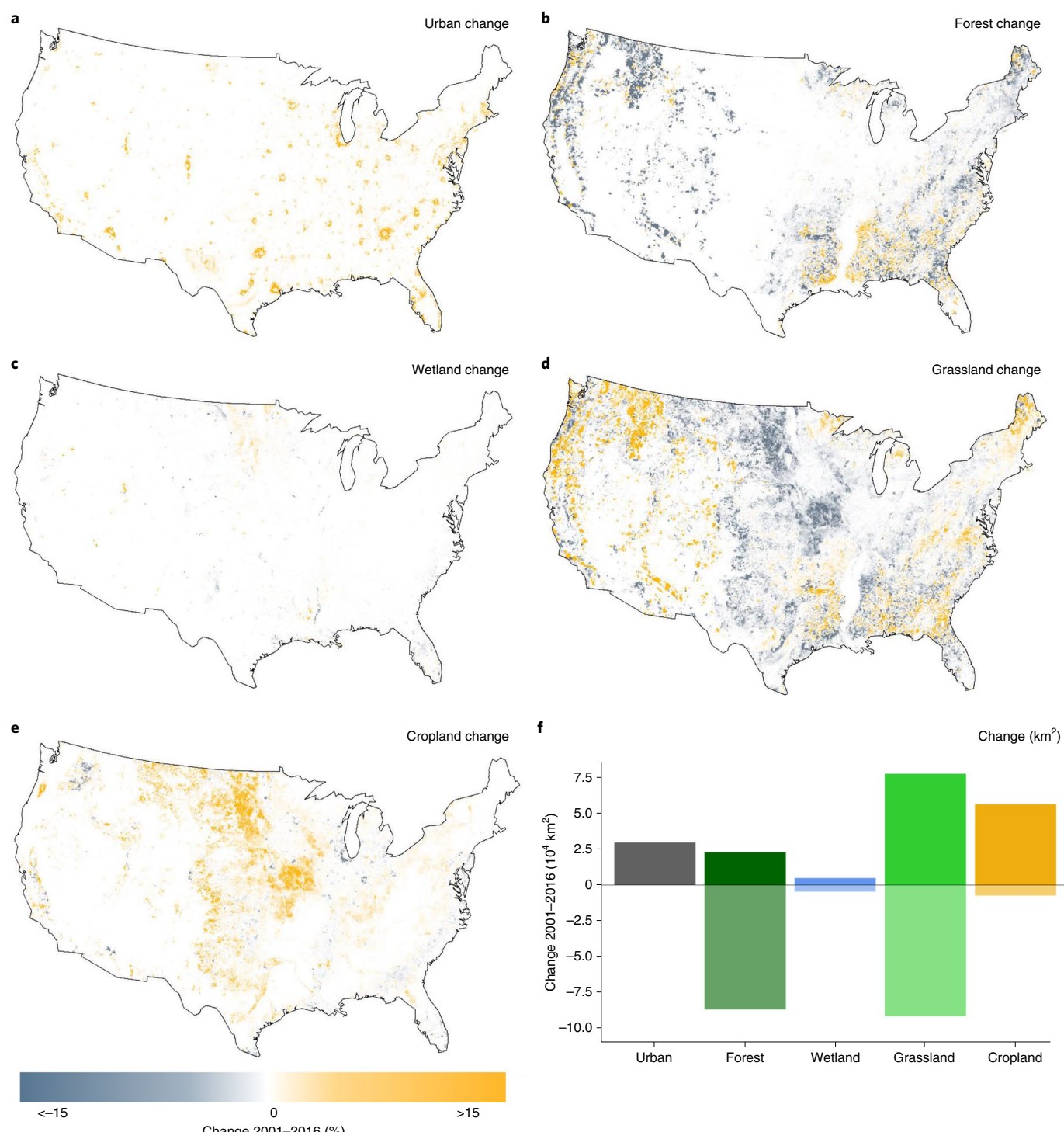

**Fig. 3 | Maps of the contiguous USA showing the spatial distribution of each land cover change type included in the analysis. a–e,** Data represent the magnitude and directionality of urban (**a**), forest (**b**), wetland (**c**), grassland (**d**) and cropland (**e**) change, in percentage points, between the years 2001 and 2016. Data were sourced from the open-access NLCD CONUS products developed by the USGS[18]. **f,** Total area of negative and positive change for each land cover covariate between 2001 and 2016 in km².

around urban centres and in the Southeast US, a region that has already experienced catastrophic losses of avian diversity and abundance over the last 50 years[15]. We show that this decline is far from being over and that more avian diversity will be lost if urgent conservation actions are not put in place. However, extensive areas of the contiguous US are also predicted to gain species, particularly in the Northeast, but also in many other less populated locations that

are close to areas predicted to be in debt. Nevertheless, we acknowledge that changes in effective numbers of species provide only a coarse measure of biodiversity change and that processes specific to species or functional traits could play a substantial role in how communities respond to habitat change. We are also aware that our results are a first attempt at quantification of biodiversity credits and debts over large spatial scales, and while this is a considerable

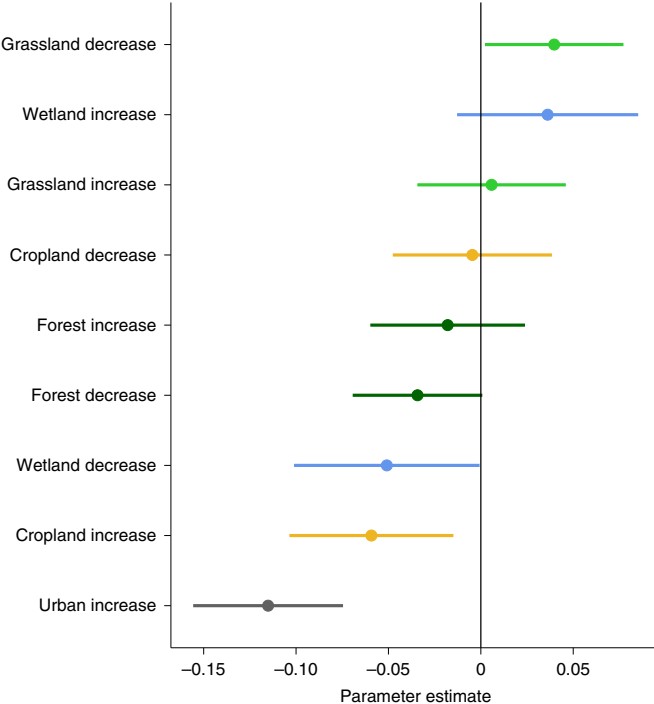

**Fig. 4 | Effect of land cover change on extinction debts and colonization credits.** Coefficient estimates (points) and credible intervals (lines) of different land cover change types of a GLM in which the response variable was the model-predicted magnitude of colonization credits and extinction debts (see Fig. 1) and the explanatory variables were the magnitudes of the positive or negative changes in the land cover types. Credible intervals are the result of uncertainty propagation by fitting the GLM to 1,000 sets of predicted values of debts and credits generated by posterior sampling. Subsequently, the parameters of each fitted GLM were sampled 1,000 times before computing the means and credible intervals presented in the figure.

improvement over assuming that equilibria are reached instantaneously, additional information could be obtained by considering multiple timepoints over a longer time period.

Taken together, our results demonstrate that extinction debts and colonization credits are widespread in avian communities across the US. This highlights the complex and dynamic nature of biodiversity responses to land use change. We argue that this complexity needs to be accounted for in predictive models to improve the projections of the impact of past, current and future habitat change on biodiversity, thus contributing to the conservation of biota worldwide.

## Methods

All of the statistical analyses were conducted using the R programming language version 4.0.5 within the RStudio IDE version 1.4.1[24,25]. Data visualization and processing were performed with the 'tidyverse' collection, 'foreach' and 'doParallel' R packages[26–28]. Geographical Information System (GIS) operations on raster and vector files were conducted using the 'sf', 'exactextractr' and 'raster' R packages[29–31].

**Data sources and pre-processing.** *Biodiversity data.* We used the North American Breeding Bird Survey dataset as our source of biodiversity data due to its long temporal coverage and spatial extent[14,32]. The BBS is composed of bird species abundance records collected since 1966 from over 4,000 survey routes across the countries of Mexico, USA and Canada. For this study we focused solely on routes in the USA, due to their longer time dimension. Data collection follows public access roads that are 24.5 miles long (circa 39.2 km) using a point count protocol whereby routes are surveyed every half-mile (800 m) for a total of 50 stops. At each stop, observers stand for 3 min and record the species and the abundance of every bird seen or heard within 400 m of their location. The routes are surveyed by volunteers with experience in bird observation, and surveys are conducted from late April to July to capture the peak of the breeding season.

We selected the years 2001 and 2016 as the two timepoints of our analysis. This 15-year timeframe corresponded to the longest possible timespan for which land cover data products were available at high spatial resolution[18]. Before analysis, we subset the BBS dataset by removing routes that had incomplete survey lengths (less than 50 point count stops, indicated by the RouteTypeDetailID field value being less than 2 in the extracted BBS dataset) or that were surveyed under adverse weather conditions such as high wind and rain (as indicated by the Run Protocol ID field being equal to 1), which could affect bird occurrence and detectability. Following this filtering process, the total number of BBS routes analysed was 960 (Extended Data Fig. 1).

For higher precision when inferring the relationship between avian diversity and environmental variables, we subdivided each route into five segments of equal length, consisting of 10 count locations each. This approach was motivated by the need to more closely associate bird communities with the land cover composition in the area in which they are found. To minimize the spatial autocorrelation between adjacent segments and avoid overlaps in landscapes analysed, we filtered the data to keep only the first, third and fifth segment of each route. These segments therefore formed our sampling unit used in all analyses.

We recognized that environmental conditions and stochastic trends in populations could introduce variability in biodiversity calculated from bird community data. We therefore extracted, for each segment and each species, the average population count across a 3-year period centred on our two timepoints (2000, 2001, 2002 and 2015, 2016, 2017)[33]. We then calculated the mean abundance of each species across these 3 years.

The effect of observer experience[34–36] was accounted for by sourcing the observer ID responsible for each route at each timepoint and including it as a random effect in the legacy model (see 'Model development' section). We also controlled for the time of day as it is plausible to expect visibility and avian species activity patterns to vary between early morning and later parts of the day. Time of day for each segment was calculated by averaging across the start and end time data entries associated with each route, and then including this as a covariate in both the legacy and equilibrium models (see 'Model development' section). However, we did not model detectability issues associated with traffic noise and disturbance for two reasons. First, all BBS surveys are conducted along public access roads with a vehicle, so the disturbance is expected to be similar across sites. Second, previous studies have found no clear evidence for noise being the main cause for reduced bird abundance near roads[37].

Following these procedures, our processed BBS dataset included entries of mean abundances of each species for a total of 2,880 segments, corresponding to segment 1, 3 and 5 of 960 routes (Extended Data Figs. 1 and 2). For each segment, at each timepoint we calculated different measures of alpha diversity following the Hill numbers framework[38]. We then selected to use the effective number of species at $q=1$, calculated as the exponential of the Shannon–Wiener Index[38]. The effective number of species at $q=1$ sits at the theoretical half-way point between the classic species richness measure that accounts only for the absolute number of species ($q=0$) and the Berger–Parker dominance index ($q=$infinity), which instead only reflects the most common species. Thus, the effective number of species is a robust alternative to species richness, which does not take account of species rarity or detectability and can thus lead to biased biodiversity estimates[16,17].

*Land cover and environmental data.* Land cover data for the US for our focal years of 2001 and 2016 were sourced from the open-access NLCD CONUS products developed by the US Geological Survey (USGS)[18,39]. The NLCD products are high-resolution (30 m pixel dimensions) classified raster files covering the land area of the whole USA. This dataset provides us with the opportunity to look at finely gridded spatio-temporal changes in a landscape over a relatively long timeframe of 15 years, while utilizing data collected and analysed with the same methods (for example, land use classification algorithms).

To reduce the number of potentially collinear explanatory variables included in our models, we aggregated the land cover variables provided by the NLCD dataset. We summarized these to five land cover categories: 'urban' (an aggregate of the Developed-Open Space (subclass 21), Developed-Low Intensity (22), Developed-Medium Intensity (23) and Developed-High Intensity classes); 'forest' (an aggregate of the Deciduous Forest (41), Evergreen Forest (42) and Mixed Forest (43) classes); 'grassland' (an aggregate of the Shrub (52), Grassland/Herbaceous (71) and Pasture/Hay (81) classes); 'cropland' (cultivated Crops (82) subclass) and 'wetland' (an aggregate of the Woody Wetland (90) and Herbaceous Wetland (95) classes). The Perennial Ice/Snow (12), Open Water (11) and Barren Land (31) classes were excluded from the analysis as they were very uncommon in our dataset. The distribution and total area of the land cover categories across the US are shown in Supplementary Figs. 1 and 2. Temperature data were sourced from the 30 arc-seconds gridded PRISM climate database[19] and were extracted as the mean across May and June for each group of years from which bird abundances were taken.

We first sampled the landscape surrounding each segment using a range of buffer shapes and sizes, and then selected the buffer type on the basis of the capacity of each buffer type to explain the response variable. The types of buffers that we explored were: a circular buffer around the centroid of the polygon defined by the

vertices of each segment (4,000 m radius) and a series of three buffers around the segment line (500 m, 2,000 m and 4,000 m radius). The best fit was given by the smallest buffer size of 500 m, shown in Extended Data Fig. 2, which also coincides with the BBS protocol effective counting distance of 400 m and more closely reflects the size of bird territories[14]. Land cover variables were computed as a percentage of the total buffer area. Change in percentage points for each land cover type between the 2 years was computed by subtracting the values at the two timepoints. A change product is also provided by the USGS databases[40], but it does not meet our needs because it considers land cover changes based on a ranking. Nonetheless, a comparison of urban land cover change between the timepoints showed a similar result (Supplementary Fig. 4). Land cover data were processed geospatially using the NAD 83 Conus Albers Coordinate Reference Systems projection, EPSG 5070.

**Model development.** *Theoretical background.* We developed a statistical model that conceptualized extinction debts and colonization credits by combining the following two concepts: (1) the settled biodiversity of avian communities in a given landscape composition (that is, a system at equilibrium) and (2) the lagged response in the species diversity in a given landscape due to recent land cover changes (that is, a system moving to a new equilibrium). We reasoned that, given enough time, and with no further changes in land cover, the effective number of species at a given location would eventually equilibrate. The equilibrium distribution of the effective number of species emerges with the waning of the legacy effect of previous landscape compositions in encouraging or impeding the recruitment and survival of particular species. We did not model these ecological mechanisms directly, but instead expressed the equilibrium of the effective number of species, and the rate of approach to this equilibrium, as empirical functions of environmental covariates. It is important to keep in mind that during a finite time interval following environmental change, it is possible that our observations of effective number of species represent a system in a transitory state towards its new equilibrium. Yet, environmental changes may occur at rates that never allow the system to equilibrate. Although the equilibration processes are latent (that is, not amenable to direct observation), the combination of equilibrium and temporal legacy components into an integrated model, applied to a dataset with extensive environmental replication (due to spatial expansiveness), has allowed us to retrieve distributions for all relevant model parameters (see below).

*Model overview.* The observed effective number of species $R_{s,t}$ at site $s$ in year $t$ for $t = t_1, t_2$ is modelled as a normally distributed variate with mean $\mu_{s,t}$ and standard deviation $\sigma$

$$R_{s,t} \sim N\left(\mu_{s,t}, \sigma\right) \tag{1}$$

We assume that, under landscape change, the system is in a state of flux and that the data are from observations witnessing the transition between two (possibly unattained) equilibria. The expected state of the system at any given point in time, $\mu_{s,t}$, was formulated as a mixture of past and future equilibrium distributions (that is, a weighted average of the two distributions, where the weights are given by the complementary proportions $\omega$ and $1 - \omega$)

$$\mu_{s,t} = f(x_{s,t_2};\beta)\,\omega\,(y_{s,i,z};\gamma) + f(x_{s,t_1};\beta)\,(1 - \omega\,(y_{s,i,z};\gamma)) \tag{2}$$

Here, the function $f$ describes the equilibrium distribution of the effective number of species as a function of the configuration of the local environment, captured in covariates $x_{s,t}$. The weighting function $\omega$ depends on covariates $y_{s,t}$ derived from the difference in the local land cover between $t_2 = 2016$ and $t_1 = 2001$ (that is, it is a function of the land cover change that has taken place). The mixture weights $\omega$ and $(1 - \omega)$ determine the relative importance of the two equilibrium distributions (past or current). If $\omega = 1$, the interpretation is that the new equilibrium distribution has been completely attained, and thus the current ($t_2 = 2016$) effective number of species is entirely explained by the current ($t_2 = 2016$) land cover. Conversely, if $\omega = 0$, the current effective number of species is entirely explained by the past ($t_1 = 2001$) land cover. The vectors of parameters $\beta$ and $\gamma$, presented in equation (2), are inferred from model fitting.

We also augmented equation (2) with a function $g$ of static covariates and random effects $z$ that we expect to have an impact on the effective number of species. Thus, the model comprised equilibrium components, a temporal legacy component and static covariates:

$$\mu_{s,t} = f(x_{s,t_2};\beta)\,\omega\,(y_{s,i,z};\gamma) + f(x_{s,t_1};\beta)\,(1 - \omega\,(y_{s,i,z};\gamma)) + g\,(z_s;\alpha) \tag{3}$$

in which $f(x_{s,t};\beta)$ are the equilibrium components for the two timepoints, $\omega\,(y_{s,i,z};\gamma)$ is the temporal legacy component, and $g\,(z_s;\alpha)$ is the function that captures the static covariates and random effects, with $\alpha$ being the estimated static covariates parameter effects.

*Equilibrium components.* We defined the equilibrium distribution of the effective number of species at a given timepoint as a function $f(x_{s,t};\beta)$ of land cover. This function describes the expected effective number of species at location $s$, given sufficient time for the community to adapt to the given land cover composition. We now describe this function in more detail.

The equilibrium component was formulated as a log-linear model comprising a total of $I = 5$ environmental covariates (the percentage cover of five landscape classes: urban, forest, grassland, wetland and cropland), using 2nd-order polynomial terms, captured by the coefficient $j$, to account for optima in effective number of species along each of the five environmental dimensions:

$$f(x_{s,t}) = \exp\left(\beta_0 + \sum_{i=1}^{I=5}\sum_{j=1}^{J=2}\beta_{i,j}x_{i,s,t}^{j}\right) \tag{4}$$

In equation (4), the $\beta$ parameters capture the effect of covariates on the equilibrium and are assumed to be the same for each environmental composition. A simplifying assumption necessary for the application of this model is that the effective number of species had equilibrated at the first timepoint. As data become available for more years in the future, the influence of this assumption on the model results will diminish and more accuracy will be achievable with multiple timepoints.

To allow for conditionality in the effects of one land cover variable on the response of the effective number of species to another land cover variable, we extended this function with pairwise interaction terms $k$ between all the linear terms for land cover variables and pairwise linear-quadratic terms, as follows:

$$f(x_{s,t}) =$$

$$\exp\left(\beta_0 + \sum_{i=1}^{I=5}\beta_{1,i}x_{s,i,t} + \sum_{i=1}^{I=5}\sum_{k=i}^{K=5}\beta_{2,i,k}x_{s,i,t}x_{k,s,t} + \sum_{i=1}^{I=5}\sum_{\substack{k=i \\ k\neq i}}^{K=5}\beta_{3,i,k}x_{s,i,t}x_{k,s,t}^{2}\right) \tag{5}$$

*Temporal legacy component.* The main covariates, $y_{s,i,z}$, for the part of the model that captures the temporal legacy, $\omega\,(y_{s,i,z};\gamma)$, are derived from the change in land cover ($\Delta x_{s,i} = x_{s,i,t_2} - x_{s,i,t_1}$) between the two timepoints

$$y_{s,i,z} = \begin{cases} y_{s,i,1} = |\Delta x_{s,i}|\,, \; y_{s,i,2} = 0, & \text{if } \Delta x_{s,i} < 0 \\ y_{s,i,1} = 0, \qquad y_{s,i,2} = \Delta x_{s,i}, & \text{otherwise} \end{cases} \tag{6}$$

where $\Delta x_{s,i,z}$ is a vector at site $s$ of the $i$th environmental change variable (that is, urban, forest, grassland, wetland, cropland) and for directionality $z$. The effect of these covariates on the mixture weights is given by:

$$\omega\,(y_{s,i,z};\gamma) = \exp\left(\sum_{i=1}^{I=5}\sum_{z=1}^{Z=2} - \gamma_{i,z}y_{s,i,z}\right) \tag{7}$$

This formulation weights the contribution that the environmental variables at the two timepoints have on the current effective number of species, as a function of the magnitude and directionality of change in each type of land cover covariate. The $\gamma$ parameters, and subsequently the temporal legacy component, are allowed via the inclusion of the environmental change data $y_{s,i,z}$, to account for the distance between the land cover at the two timepoints, therefore quantifying how far the initial community would need to travel to reach equilibrium in 2016 as a function of the type, magnitude and directionality of change. It should be noted that our model, in equation (3), is only implicitly related to the speed with which the effective number of species reacts to environmental changes. Instead, it quantifies how much further it would still have to travel to reach the expected equilibrium associated with the current configuration of the landscape.

*Static covariates.* As described in model equation (3), we included a function of static covariates to which we can expect the effective number of species to respond without lags relating to the past landscape. We added a linear and quadratic fixed effect for temperature in 2016 to control for any difference in the effective number of species related to climatic characteristics and to allow for a parabolic relationship to be expressed (optima either at mean or extremes values). We also controlled for the heterogeneity of a landscape by including the effective number of land cover types, computed in the same way as the effective number of species, as a fixed effect[40]. A fixed effect for time of day, reflecting the time at which each segment was surveyed, was included to correct for differences in species detectability between early morning and later parts of the day[41]. An observer-level random effect was also added to control for variation between observers[35,36] and partly account for between-route variation, given that we would expect observers who collect data from multiple routes to do so within a relatively small area. Spatial autocorrelation of the effective number of species was tested for all segments at once and by different radiuses for neighbour inclusion (500 m, 1,000 m, 5,000 m, 10,000 m, 100,000 m), using the Moran's I statistic[42]. Spatial autocorrelation was not corrected for because Moran's I was not significant at any spatial scale ($P > 0.05$). Pseudo-replication between neighbouring segments was avoided by considering segments 1, 3 and 5, whose land cover buffers did not overlap (Extended Data Fig. 2).

**Model fitting.** The model was fitted within a Bayesian framework using a Hamiltonian Markov chain Monte Carlo algorithm implemented in the STAN programming language[43] version 4.3.0 and the 'cmdstanr' R package version 2.26.1[44].

We ran 4 chains, sampling for 1,000 iterations with a burn-in period of 500 iterations each. These numbers of iterations were sufficient to achieve chain convergence. The STAN sampling was run on four parallel threads on a multi-core Intel i7 – 8750H processor with a maximum clock speed of 4.1 GHz.

For the purposes of Bayesian inference, all slope parameters associated with the equilibrium component equation (5) and the static additive terms were assigned an unbiased prior $\beta_{i,j} \sim N(0,1)$ and $z_s \sim N(0,1)$, where $N$ is normal, with the aim of shrinking the parameter estimated towards 0 (that is, no covariate effect). A gamma distributed prior, with shape and rate 0.001, was assigned to the standard deviation of the random effect. For the following known and expected relationships, we also truncated the range of parameter values by bounding the upper or lower limits of the prior/posterior distributions. Intercept and standard deviation of the observer random effect were bounded below by 0. Linear effects for the environmental covariates and temperature were bounded below at 0, while their quadratic counterparts were bounded above at 0. Interaction terms were not limited. The temporal legacy component parameters were given a uniform ($U$) prior $\gamma_i \sim U(0,1)$, bounded between 0 and 1 to act as a weighting proportion between the present and the past. The upper bound on the gamma parameters to 1 does not bias us towards an increased contribution of the past land cover, but instead provides a more conservative approach.

Model diagnostics were conducted by assessing chain convergence visually through trace plots, as well as statistically by employing the Gelman-Rubin test, which compares the estimated between-chain and within-chain variances[45]. Chain autocorrelation and the associated effective sample size were also monitored. In the case of low effective sample size, the chains were extended until the effective sample size exceeded a threshold value of 400. The marginal posterior distribution for each parameter was visualized via a density plot to check for multimodality.

Model selection was conducted to inform choice of the size and shape of the land cover buffer around each sampled segment. We did so by comparing values of the Watanabe-Akaike Information Criterion leave-one-out (WAIC)-loo information criterion[46] of four different models, each computed using land cover data calculated with two different buffer options of various sizes: a circular buffer around the centroid of the polygon defined by the vertices of each segment (4,000 m radius) and a series of buffers around the segment line (500 m, 2,000 m and 4,000 m radius). This approach was implemented through the 'loo' R package version 2.1, which provides an improvement on the original WAIC by including diagnostic measures around the point-wise log-likelihood value estimated around each sample draw[47].

**Visualization of model predictions.** A map of the USA (Fig. 1) was produced to represent the predicted extinction debts and colonization credits (that is, positive or negative distance in the effective number of species from the expected equilibria). The map was produced on a hexagonal grid at a spatial resolution of 10 km vertex-to-opposite-vertex, with each hexagon covering a total of 86 km². Values of extinction debt and colonization credit were calculated by subtracting the predicted effective number of species produced by the model (equation 3) from the predicted effective number of species at equilibrium in 2016 (that is, when the legacy component equals 1). To correctly propagate and represent uncertainty in the extinction debts and colonization credits presented, this process was repeated 1,000 times for predictions originating from different draws from the posterior distribution. Uncertainty in the form of the geometric coefficient of variation, calculated as $\sqrt[2]{e^{(\log(\sigma+1)^2)}} - 1$ where $\sigma$ is the standard deviation, is mapped in Extended Data Fig. 4a. Extended Data Fig. 4 also includes a copy of Fig. 1 (Extended Data Fig. 4b) for reference, alongside upper (Extended Data Fig. 4c) and lower (Extended Data Fig. 4d) credible intervals.

Over/underestimation values of biodiversity that could arise by neglecting debts and credits were computed as the difference between the effective numbers of species predicted by the equilibrium model and the legacy model, multiplied by 100 and then divided by the predicted effective number of species under the legacy model. This calculation results in a percentage measurement of the extent to which (in relative terms) the current effective number of species under- or overestimates the diversity that a given landscape can sustain at equilibrium.

To further validate our predicted extinction debts and colonization credits, we compared the direction of the expected changes with the recorded difference in effective numbers of species between 2016 and 2019 (the latest year for which data are available). To do so, we sourced bird abundances from the North American BBS dataset[14,32] for the year 2019 and conducted the same data processing as described above for the other two timepoints. We then conducted a Pearson correlation test to assess how well the observed change followed the model-predicted one. We are nevertheless aware that a 3-year timeframe is unlikely to be large enough for debts and credits to fully manifest.

Plots were also generated to describe the behaviour of the mixture weight, $\omega$ (equation 7), which captures the contribution (weighting) of the landscape composition in determining the effective numbers of species at the two timepoints (Fig. 2 in the main text). Values of $\omega$ across the whole spectrum of plausible land cover change values (that is, from −100 to +100) were simulated by averaging

over 10,000 draws from the posterior distribution of each $\gamma$ parameter. Credible intervals were measured by taking the 95% range of the 10,000 draws.

**Explaining spatial variation in debts and credits.** The extinction debts and colonization credits predicted for the contiguous USA states were further modelled to identify which past land cover changes were the main drivers of the delayed biodiversity changes in USA bird communities. We considered the values of debts or credits associated with the 92,000 individual 86 km² hexagons (Fig. 1) as a response variable. We then specified a Gaussian linear model including the magnitude of each land cover change as explanatory covariates. Positive and negative changes in each covariate were treated as separate linear components to differentiate their effects. The model was fitted to 1,000 sets of debts and credits, each originating from predictions based on independent draws from the posterior distribution. For each generalized linear model (GLM) fit, we then subsequently sampled each parameter distribution another 1,000 times and extracted the summarized parameter estimates from a total of 100,000 values. Model coefficients and their resulting uncertainty from the above process are presented in Fig. 4 and in more detail as part of Supplementary Table 3.

**Reporting Summary.** Further information on research design is available in the Nature Research Reporting Summary linked to this article.

## Data availability

All data utilized in the analysis are open access. Data on bird abundances can be accessed at https://www.pwrc.usgs.gov/BBS/RawData/. Data on the land cover and temperature covariates can be accessed at https://www.mrlc.gov/ (land cover) and https://prism.oregonstate.edu/ (temperature). BBS routes were sourced from https://databasin.org/datasets/02fe0ebbb1b04111b0ba1579b89b7420/.

## Code availability

Reproducible R code and processed datasets are available from https://github.com/valiriel/USBBS_Biodiversity_LandCover_Delays.

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

## Acknowledgements

We thank fellow academics in the Institute of Biodiversity, Animal Health and Comparative Medicine at the University of Glasgow for the support and feedback received during the paper finalization process. This project was supported financially by the University of Glasgow. D.M.D. was funded by a grant from the UK Natural Environment Research Council (NERC) (NE/S005773/1). R.M.'s contribution to this work was supported by The Leckie Fellowship, the UK Medical Research Council (grant number MC_UU_00022/4) and the Chief Scientist Office (CSO) (grant number SPHSU19) at the MRC/CSO Social and Public Health Sciences Unit, University of Glasgow.

## Author contributions

Y.H., R.M., J.M., S.S. and D.M.D. conceived and designed the study. Y.H., R.M. and J.M. performed all the computational analyses. Y.H., R.M., J.M., S.S. and D.M.D. wrote the manuscript.

## Competing interests

The authors declare no competing interests.

## Additional information

**Extended data** is available for this paper at https://doi.org/10.1038/s41559-021-01653-3.

**Correspondence and requests for materials** should be addressed to Davide M. Dominoni.

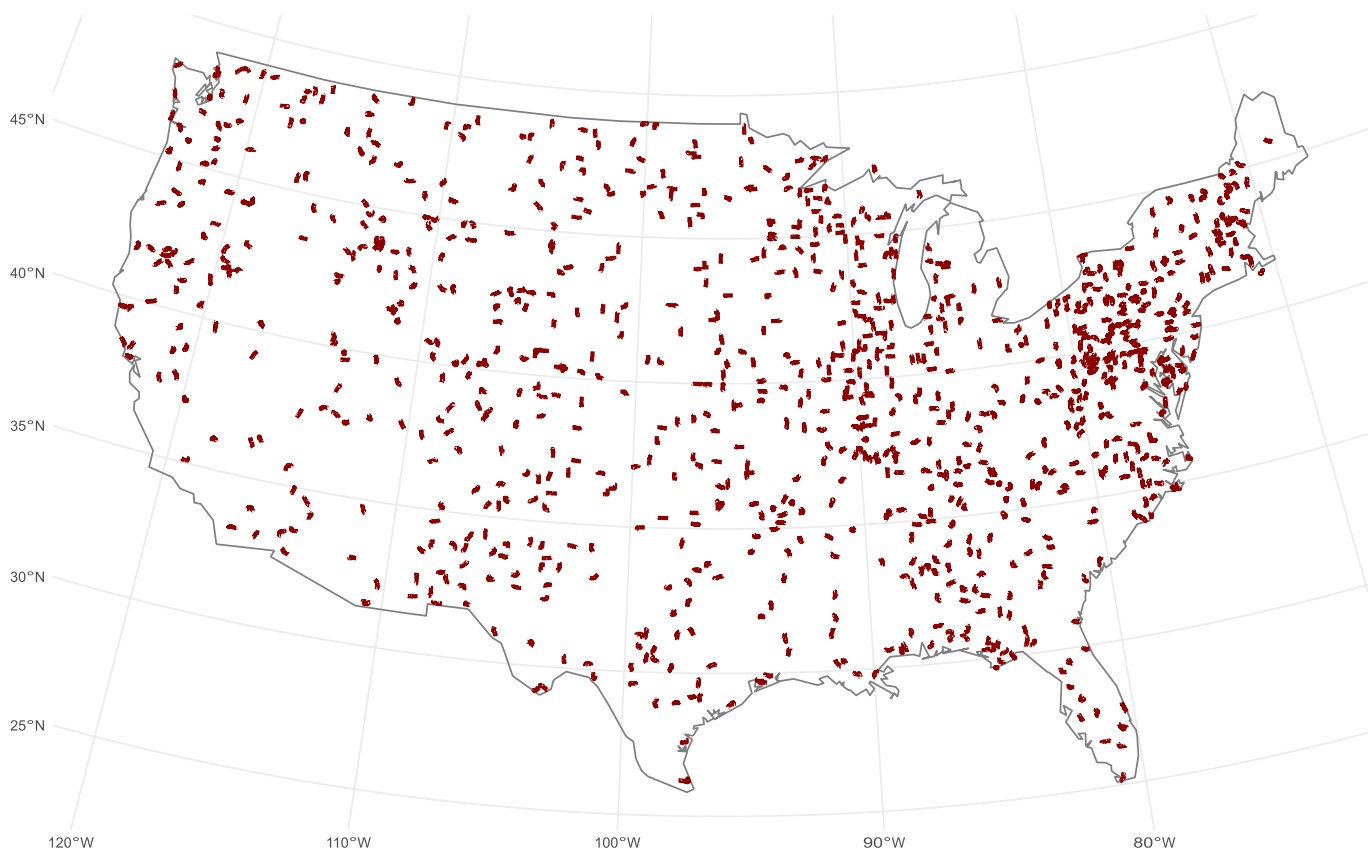

**Extended Data Fig. 1 | BBS routes.** Distribution of the 960 analysed routes of the United States Breeding Bird Survey across the contiguous USA. This represents a subset of routes which were consistently surveyed across the two timepoints of interest and surrounding years (2000, **2001**, 2002 and 2015, **2016**, 2017).

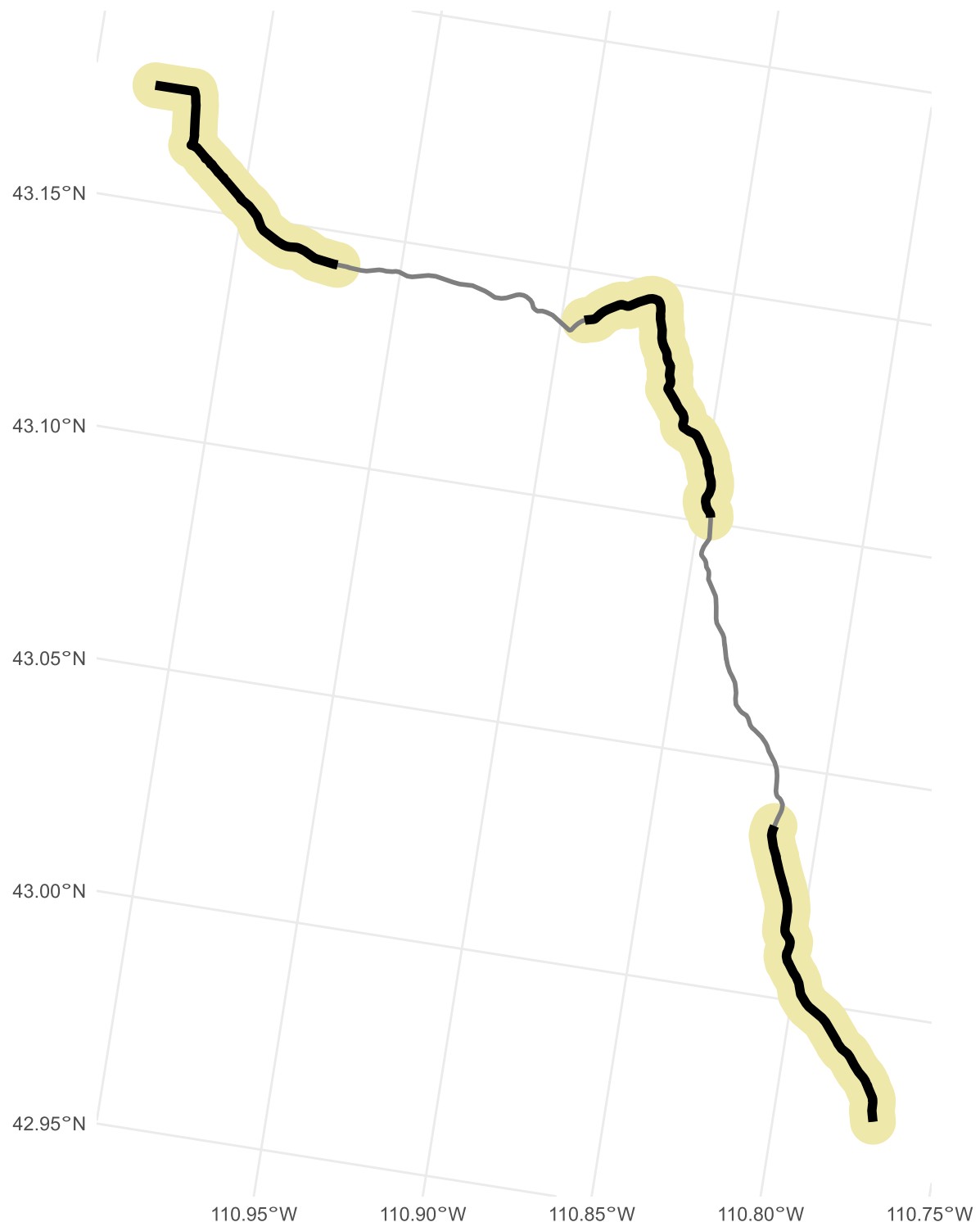

**Extended Data Fig. 2 | Route buffers.** Visual representation of a Breeding Bird Survey route segments and spatial buffers. Each route is approximately 40 km long. Visible are five segments, each representing 10 bird point counts. In bold black are segment one, three and five, which were the source of biodiversity and landscape data used in the model. The buffer from which landscape metrics were sampled is shown in pale yellow. Buffers are 500 metres distant from each segment line, and buffer size was selected by comparing model fit between several shapes and sizes. In grey, are segment two and four, which we excluded from the analysis to minimise pseudo-replication that would otherwise arise from the proximity of the route segments.

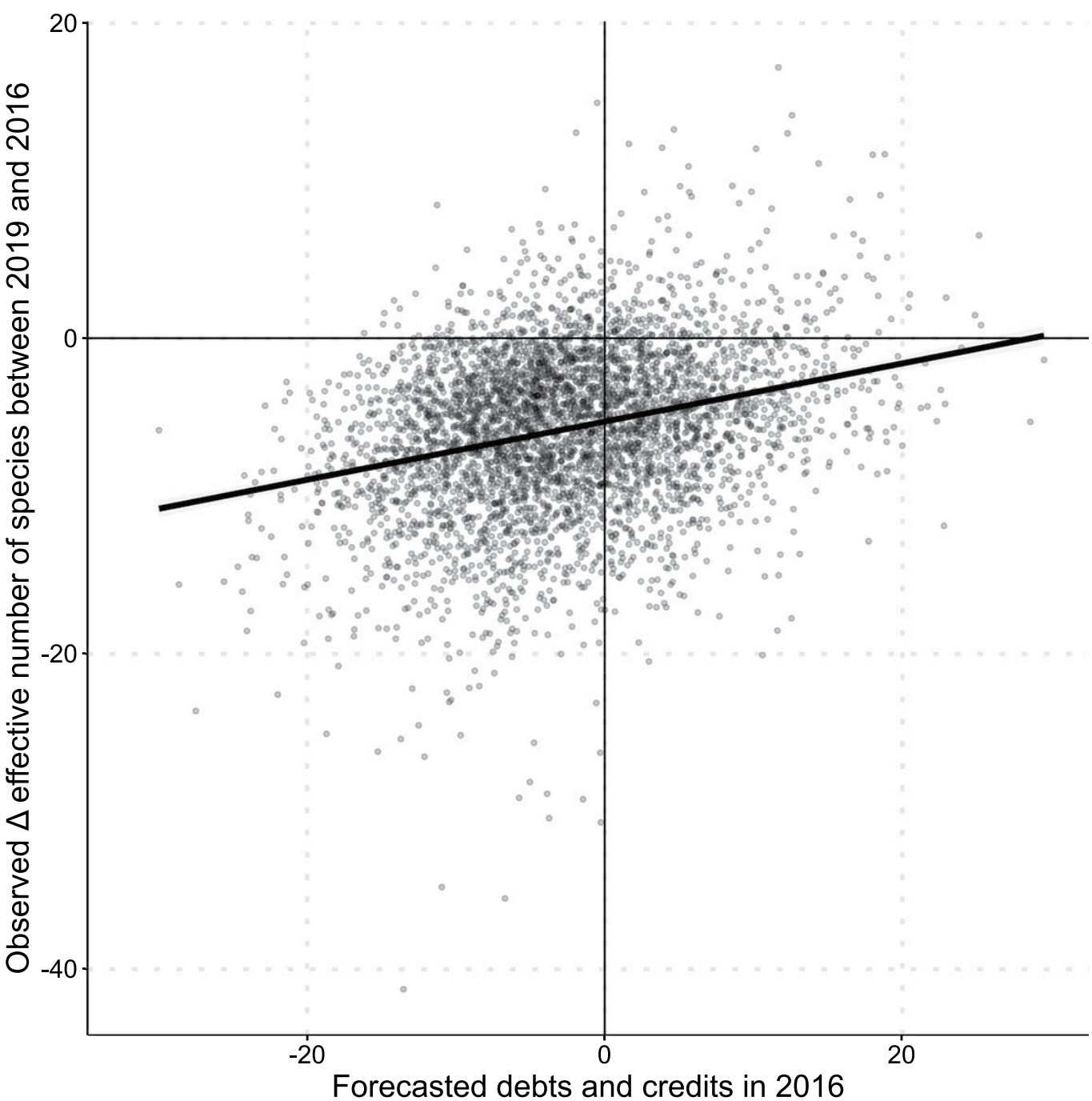

**Extended Data Fig. 3 | Model validation.** Correlation between the observed change in effective number of species between 2016 and 2019 and the model-predicted values of extinction debt and colonisation credits. Data was obtained from 4233 US bird communities (subset of the 4800 communities with data also available in 2019). Despite the relatively short time interval (we expect most of these debts and credits will require longer before they can be fully realized), changes in effective number of species since 2016 have overall been in the direction predicted by our model (Pearson correlation test, r = 0.28, df = 4233, p < 0.001).

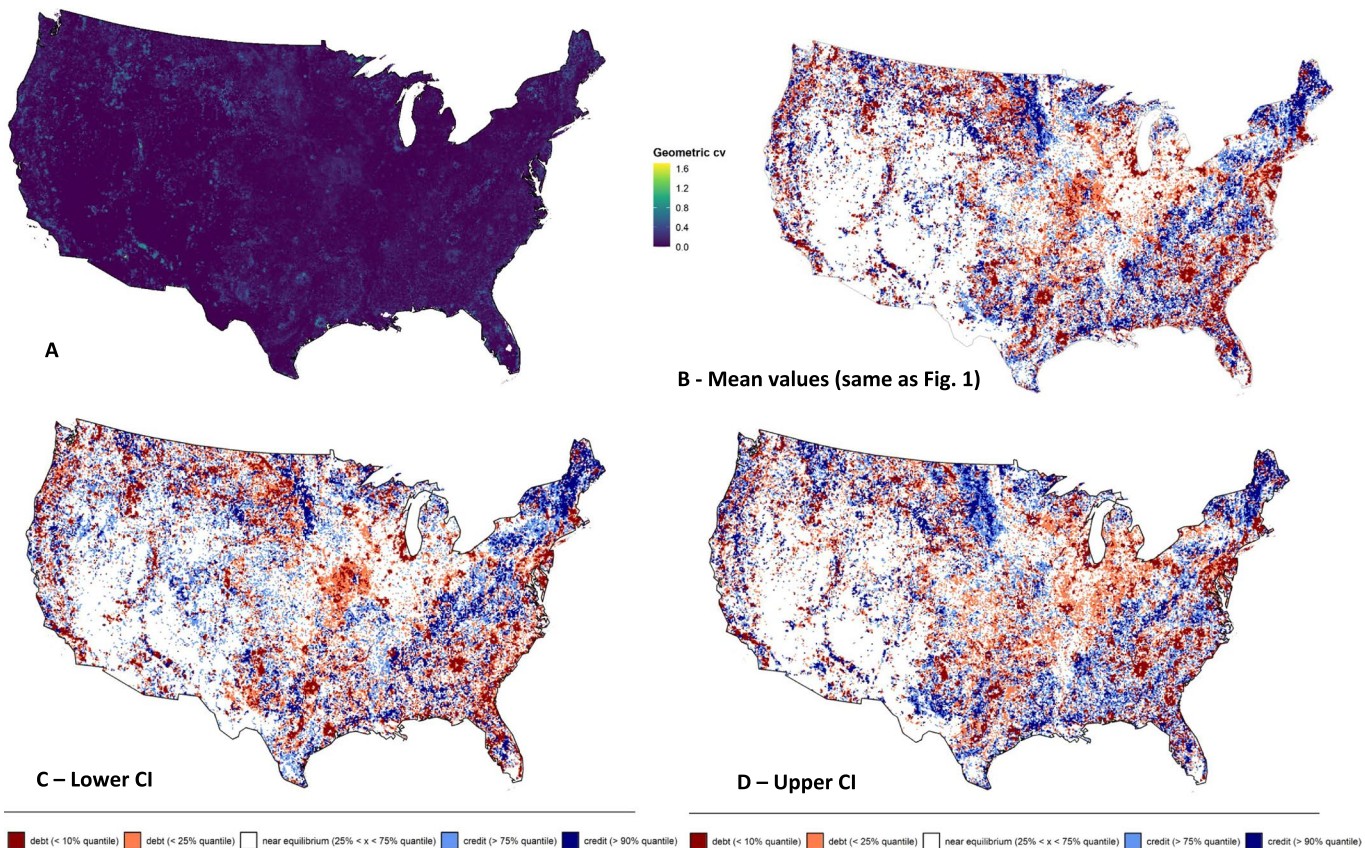

**A**

**B - Mean values (same as Fig. 1)**

**C – Lower CI**

**D – Upper CI**

Geometric cv
1.6
1.2
0.8
0.4
0.0

■ debt (< 10% quantile)  ■ debt (< 25% quantile)  □ near equilibrium (25% < x < 75% quantile)  ■ credit (> 75% quantile)  ■ credit (> 90% quantile)    ■ debt (< 10% quantile)  ■ debt (< 25% quantile)  □ near equilibrium (25% < x < 75% quantile)  ■ credit (> 75% quantile)  ■ credit (> 90% quantile)

**Extended Data Fig. 4 | Mapping model uncertainties.** Maps of the contiguous USA showing the uncertainty around the predictions of extinction debts and colonisation credits. We first sampled from the posterior distributions of the equilibrium and legacy models, and computed the difference between the predicted values of the effective number of species of both models. The process was repeated 1000 times for each of the circa 92,000 landscape compositions mapped in panel B (same as Fig. 1 in main text). We then calculated the geometric coefficient of variation of these predicted values predicted and presented it in panel A. Panels C and D show, respectively, the upper (97.5%) and lower (2.5%) credible intervals of the predicted values.

# Reporting Summary

## Statistics

For all statistical analyses, confirm that the following items are present in the figure legend, table legend, main text, or Methods section.

| n/a | Confirmed | |
|---|---|---|
| ☐ | ☒ | The exact sample size (*n*) for each experimental group/condition, given as a discrete number and unit of measurement |
| ☐ | ☒ | A statement on whether measurements were taken from distinct samples or whether the same sample was measured repeatedly |
| ☐ | ☒ | The statistical test(s) used AND whether they are one- or two-sided *Only common tests should be described solely by name; describe more complex techniques in the Methods section.* |
| ☐ | ☒ | A description of all covariates tested |
| ☐ | ☒ | A description of any assumptions or corrections, such as tests of normality and adjustment for multiple comparisons |
| ☐ | ☒ | A full description of the statistical parameters including central tendency (e.g. means) or other basic estimates (e.g. regression coefficient) AND variation (e.g. standard deviation) or associated estimates of uncertainty (e.g. confidence intervals) |
| ☐ | ☒ | For null hypothesis testing, the test statistic (e.g. *F*, *t*, *r*) with confidence intervals, effect sizes, degrees of freedom and *P* value noted *Give P values as exact values whenever suitable.* |
| ☐ | ☒ | For Bayesian analysis, information on the choice of priors and Markov chain Monte Carlo settings |
| ☐ | ☒ | For hierarchical and complex designs, identification of the appropriate level for tests and full reporting of outcomes |
| ☒ | ☐ | Estimates of effect sizes (e.g. Cohen's *d*, Pearson's *r*), indicating how they were calculated |

*Our web collection on statistics for biologists contains articles on many of the points above.*

## Software and code

Policy information about availability of computer code

| Data collection | RAll data utilised in the analysis is open access. Data on bird abundances can be accessed at: https://www.pwrc.usgs.gov/BBS/RawData/. Data on the land cover and temperature covariates can be accessed at: https://www.mrlc.gov/ (land cover) and https://prism.oregonstate.edu/ (temperature). BBS routes were sourced from https://databasin.org/datasets/02fe0ebbb 1b04111b0ba1579b89b7420/ |
|---|---|
| Data analysis | Reproducible R code and processed datasets are available from https://github.com/valiriel/USBBS_Biodiversity_LandCover_Delays. |

For manuscripts utilizing custom algorithms or software that are central to the research but not yet described in published literature, software must be made available to editors and reviewers. We strongly encourage code deposition in a community repository (e.g. GitHub). See the Nature Portfolio guidelines for submitting code & software for further information.

## Data

Policy information about availability of data

All manuscripts must include a data availability statement. This statement should provide the following information, where applicable:
- Accession codes, unique identifiers, or web links for publicly available datasets
- A description of any restrictions on data availability
- For clinical datasets or third party data, please ensure that the statement adheres to our policy

All data utilised in the analysis is open access. Data on bird abundances can be accessed at: https://www.pwrc.usgs.gov/BBS/RawData/. Data on the land cover and temperature covariates can be accessed at: https://www.mrlc.gov/ (land cover) and https://prism.oregonstate.edu/ (temperature). BBS routes were sourced from https://databasin.org/datasets/02fe0ebbb 1b04111b0ba1579b89b7420/

# Field-specific reporting

Please select the one below that is the best fit for your research. If you are not sure, read the appropriate sections before making your selection.

☐ Life sciences        ☐ Behavioural & social sciences        ☒ Ecological, evolutionary & environmental sciences

For a reference copy of the document with all sections, see nature.com/documents/nr-reporting-summary-flat.pdf

# Ecological, evolutionary & environmental sciences study design

All studies must disclose on these points even when the disclosure is negative.

| | |
|---|---|
| Study description | We used species richness data from the North American Breeding Bird Survey (BBS), comprising information on the abundance of 541 bird species across the contiguous USA. We also sourced high spatial resolution (30m2) land cover data from the National Land Cover Database CONUS products, as well as temperature data (mean across May and July) from the PRISM climate dataset. Using these datasets, we developed a generalized mixed effects model (GLMM) within a Bayesian framework describing the number of species in 2016 as a function of the weighted contribution of landscape composition in 2001 and 2016. This enabled us to explicitly quantify the importance of legacy effects in the response of bird communities to % changes in each of five major land cover classes (forest, grassland, cropland, wetland and urban area). |
| Research sample | We used the North American Breeding Bird Survey (BBS) dataset as our source of biodiversity data due to its long temporal coverage and spatial extent. The BBS is composed of bird species abundance records collected since 1966 from over 4000 survey routes across the countries of Mexico, USA and Canada. |
| Sampling strategy | Our final dataset included species richness and evenness data for 960 routes, each divided into five segments, giving a total of 2880 observational units (that we refer to as "segments"). |
| Data collection | Data collection follows public access roads along non-linear transects that are 24.5 miles long (circa 39.2 Km) using a point count protocol whereby routes are surveyed every half-mile (800 m) for a total of 50 stops. At each stop, observers stand for three minutes and record the species and the abundance of every bird seen or heard within 400 meters of their location. The routes are surveyed by volunteers with experience in bird observation, and surveys are conducted during May and July to capture the peak breeding season. |
| Timing and spatial scale | To address our research questions, we selected the years 2001 and 2016 as our two timepoints. This 15-year timeframe was selected as a reasonable scale to explore biodiversity lags to land cover change and it also corresponded to the longest possible timespan for which land cover data products were available at high spatial resolution. To minimise the noise in bird community data associated with stochastic annual variability in environmental conditions, we selected, for each sampling point and each species, the average population count across three adjacent years (2000, 2001, 2002; 2015, 2016, 2017) |
| Data exclusions | For this study we focused solely on routes in the USA, as most Mexican and Canadian routes are currently still being set up, therefore data in these regions are spatially and temporally sparse. Prior to analysis, we filtered the BBS dataset by removing routes that had incomplete survey lengths (less than 50 point count stops, indicated by the RouteTypeDetailID field value being less than 2 in the extracted BBS dataset), routes that were surveyed under adverse weather conditions such as high wind and rain (as indicated by the Run Protocol ID field being equal to 1), which could affect bird occurrence and detectability. We also removed segments 2 and 4 from our analyses, thus considering only segments 1-3-5, to minimise spatial autocorrelation. |
| Reproducibility | Since this is a modelling study using freely available data, and we made our code available to the community, the study can be reproduced by anybody. |
| Randomization | Sampling was not random as data collection depends on volunteers, and it's therefore dependent on population density. We have repeated our analyses with a subselection of the data to have a more equally spatially-distributed dataset, but this did not change the results. |
| Blinding | We have used all bird data from the USA irrespective of who collected it and where it was collected. |

Did the study involve field work?    ☐ Yes    ☒ No

# Reporting for specific materials, systems and methods

We require information from authors about some types of materials, experimental systems and methods used in many studies. Here, indicate whether each material, system or method listed is relevant to your study. If you are not sure if a list item applies to your research, read the appropriate section before selecting a response.

## Materials & experimental systems

| n/a | Involved in the study |
|-----|----------------------|
| ☒ | Antibodies |
| ☒ | Eukaryotic cell lines |
| ☒ | Palaeontology and archaeology |
| ☒ | Animals and other organisms |
| ☒ | Human research participants |
| ☒ | Clinical data |
| ☒ | Dual use research of concern |

## Methods

| n/a | Involved in the study |
|-----|----------------------|
| ☒ | ChIP-seq |
| ☒ | Flow cytometry |
| ☒ | MRI-based neuroimaging |

