## [Peer Review File · Nature Ecology & Evolution]

Peer Review Information

Journal: Nature Ecology & Evolution

Manuscript Title: Invisible biodiversity: widespread extinction debts and colonisation credits in United States breeding bird communities

Corresponding author name(s): Davide M. Dominoni

Reviewer Comments & Decisions:

Decision Letter, initial version:
--

18th February 2021

Dear Dr Dominoni,

Your Article, "Invisible biodiversity: A longitudinal gradient of extinction debts and colonization credits in US bird communities" has now been seen by three reviewers, whose comments are copied below. Although the reviewers find your work timely and potentially important, in the light of their overall advice I regret to say that we are unable to offer to publish it in Nature Ecology and Evolution, at least in its present form.

You will see from their reports that Reviewers 2 and 3 feel that your manuscript has potential, but Reviewer 1 raises numerous issues with the use of the Breeding Bird Survey Data. The extent of these comments suggests that the data processing and analyses would need to be overhauled before the conclusions could be deemed robust.

Further, Reviewer 1 and 3 both raise concerns about the uncertainty in the modeling and the extent to which the model predictors explain much absolute variation in the data. It seems that a significant amount of additional analyses will be required to address this uncertainty, and it is far from clear whether the results will continue to appear impressive in the light of this additional work. And while we would not rule out consideration of a revised manuscript that makes a much stronger case in support of the claims, we feel that, at this stage, the present work is at too preliminary a stage to warrant publication in Nature Ecology and Evolution.

Whether or not it will be possible for you to address all of these concerns is not something we can assess at this stage; we would be reluctant to trouble the reviewers again unless we thought that their

comments had been fully addressed. If you find that you can address the concerns, you may submit a revision along with other information as described below.

* Include a "Response to reviewers" document detailing, point-by-point, how you addressed each reviewer comment. If no action was taken to address a point, you must provide a compelling argument. This response will be sent back to the reviewer along with the revised manuscript.

* If you have not done so already we suggest that you begin to revise your manuscript so that it conforms to our Article format instructions at <http://www.nature.com/natecolevol/info/final-submission>. Refer also to any guidelines provided in this letter.

* Include a revised version of any required reporting checklist. It will be available to reviewers (and, potentially, statisticians) to aid in their evaluation if the manuscript goes back for peer review. A revised checklist is essential for re-review of the paper.

[REDACTED]

If you wish to submit a suitably revised manuscript we would hope to receive it within 6 months. If you cannot send it within this time, please let us know. We will be happy to consider your revision so long as nothing similar has been accepted for publication at Nature Ecology & Evolution or published elsewhere.

Nature Ecology & Evolution is committed to improving transparency in authorship. As part of our efforts in this direction, we are now requesting that all authors identified as 'corresponding author' on published papers create and link their Open Researcher and Contributor Identifier (ORCID) with their account on the Manuscript Tracking System (MTS), prior to acceptance. This applies to primary research papers only. ORCID helps the scientific community achieve unambiguous attribution of all scholarly contributions. You can create and link your ORCID from the home page of the MTS by clicking on 'Modify my Springer Nature account'. For more information please visit www.springernature.com/orcid.

Thank you for the opportunity to review your work.

[REDACTED]

Reviewer expertise:

Reviewer #1: Breeding Bird Survey data

Reviewer #2: Ecological lags, bird conservation

Reviewer #3: Extinction debts, Bayesian spatial analyses

Reviewers' comments:

Reviewer #1 (Remarks to the Author):

This is a very interesting manuscript, and it has some interesting ideas. But it is a risky analysis in several ways. It is a very complicated model that works with a grossly oversimplified response variable. It plays fast and loose with many of the big data sets that are used, and misuses them in some important ways. Some of the rationales for approaches are misleading. Finally, it ends with a variety of generalizations to explain the patterns that will likely not be convincing to North American ornithologists.

General comments:

1. The authors seem to not pay attention to concerns that are central to BBS analyses.
 - a. Observer differences are quite important in BBS analyses, both in estimation of population change of individual species (see Sauer et al, Kendall et al. references in this manuscript) and in species richness estimation. The text misrepresents prior research results.
 - b. Taking maximums by species over years is ad-hoc and is a flawed means of increasing detection of species, as populations are also changing between years. In general, replication over years is not a feasible strategy of accommodating issues of detectability. Evenness, as estimated from BBS counts, is not a credible metric due to varying detectability among individuals of different species.
 - c. 10-stop samples units introduce complexity into the analysis. It introduces spatial uncertainty, as no data exists on where the stops are along the route (Trust me on this; I authored the route path information used in this analysis). That error propagates along the route, if you split up the route geographic information by starting at one end and measuring along the route path.
 - d. Another complexity in the 10-stop sample unit is that BBS routes are always run sequentially from stop 1 to stop 50. That means the first 10 are early (before or at sunrise), and the last 10 are much later in the morning (3 hours after sunrise). You include a route effect, but not a time-of-day effect.
 - e. There is an extensive literature on estimation of species richness from BBS data (see papers by Nichols et al. and Boulinier et al), and there are a variety of potential multispecies occupancy approaches for analysis of BBS species richness. The authors neither cite or consider the use these methods, even though they provide a much better alternative to ad-hoc multiyear summaries.
2. Total species richness strikes me as an insensitive metric when clearly different species groups respond differently to loss or gain of different habitats.
3. Land use and land cover data from USGS have historically not been comparable between major releases due to changes in interpretations methods. USGS had addressed this issue by directly computing change based on reanalyses of prior releases. Is that a concern here? At a minimum, a citation is needed on comparability of categorizations over time
4. Scale of summary is also an issue. The circular "buffers" have several disadvantages, as points run

from center to edge, and their only virtue is that all points will be inside them. But they overlap. Perhaps that is not a great concern for explanatory variables, but it merits discussion.

Specific comments:

I. 83. What is this "novel bird data?"

I. 188. This statement is incorrect. Canada data are comparable to US data over southern Canada, so it is absolutely incorrect to say that Canada routes "are currently still being set up."

I. 190. Omit "along non-linear transects." The routes follow roads, pure and simple.

I. 194. Surveys are conducted Late April-July, not "May and July." June, of course, is the primary month of sampling.

I. 197-199. This seems backward. The 15 year timeframe is the longest possible timeframe with data cover data and also are a reasonable scale for exploring biodiversity. It is not as though you chose a 15 year interval and just happened to have data in that time frame.

I. 201. What is "a sampling point?" A route? Or a stop?

I. 202. I am always suspicious of ad-hoc approaches that use maximum counts over years "to minimize noise in bird community data." Many factors influence BBS counts, and many analyses control for observer differences. Also, population change, sometimes dramatically, from year to year. Simply taking Max values over multiple years ignores (and introduces bias in estimation) associated with these factors.

I. 209-210. Two issues here: (1) Are these 10-stop totals the quantities that were Max'ed over the 2 years (i.e., the units called "sampling points, above")". (2) How do you find the areas associated with these 10 stops? Stops locations are not documented for the BBS? Another quite important issue is that BBS routes are conducted sequentially, so visibility and other features of diurnal activity almost certainly vary among these sample units. This does not seem to be controlled for anywhere in the analysis.

I. 215. This is incorrect. At least 2 of the cited studies on observer differences DID find observer effects (I am a coauthor on them). There is clear evidence of observer differences in the BBS, and it is clear that observer quality is increasing over time. One might argue that observer differences build in increases into BBS data, but one cannot argue that observer differences are not important.

I. 218. This is the first mention of "species richness and evenness data." You need to define them in the content of your samples.

I. 221. Here is yet another term for the sample unit. Is an "observational unit" a "sampling point" and a "10 stop." Define the sampling units clearly, and use a consistent term to reference them.

I. 246-253. I have several concerns with this spatial summary. (1) The areas used to assess land use overlap along routes, causing a dependence. (2) the locations of the apparent sample units (10 stop groups) are uncertain as stop locations are not known). (3) Adjacent 10 stop units within a route are

not independent (i.e., a pseudoreplication concern unless accommodated in the analysis) Note..I see that this is included (l. 367), but should be mentioned here as a concern. (4). 3km seems like a very large area. A "dispersal distance" seems irrelevant for most migratory birds and resident birds, for different reasons. It is also much larger than the effective counting distance along BBS routes (400m), and is also much larger than species territories.

l. 260. A "bespoke statistical model?" I had to Google this. Doesn't the word "developed" make "bespoke" redundant?

l. 279-281. Why is species richness a Poisson? As the sum of a series of variables (i.e., the occurrence of the species), I would think it would be Normal.

Reviewer #2 (Remarks to the Author):

REVIEW: "Invisible biodiversity: A longitudinal gradient of extinction debts and colonization credits in US bird communities"

In this paper the authors quantify extinction debts and colonisation credits on bird communities as a result of recent (15-year) land use changes in the USA. I consider that the paper is interesting and novel within the 'ecological time lags' literature mainly because 1) it encompasses more than one land cover type, and 2) it looks at the effects habitat gains and losses (thus acknowledging that landscapes are dynamic). The authors make good use of a large dataset of bird communities (with repeated measures over two time periods) over a large geographical area. The abstract clearly communicates the key findings of the work and the figures presented in the main text are appropriate and informative. The paper is generally well-written, however, I found the language in relation to 'debts vs. credits', 'strong vs. weak lags', 'high vs. low contribution of past timepoints' rather confusing, and found myself referring to the figures (and figure legends) to try to understand what the authors meant. There also seem to be some inconsistencies in the descriptions of how credits/debts were calculated (see e.g. comments for L68 and L410). I therefore think the main text would benefit from clearer definitions and some rewording to simplify the interpretation of the findings. Additionally, I think that the discussion of the findings could be expanded to make the most of the results presented (see e.g. comments for L107 & Fig. 3) and to more prominently acknowledge that focusing on changes in species richness provides only a coarse measure of biodiversity change. Below I provide some more specific comments which I hope the authors will find useful to improve the quality of their paper.

SPECIFIC COMMENTS:

L39 – 'most' community responses to habitat change are not instantaneous. In some cases they can be, e.g. if the habitat change is particularly severe/abrupt it can extirpate rare species with very small/localised populations.

L52-53 and also L81-89 – 'validated our predictions utilizing independent data from a more recent survey' [...] 'we further validated our predictions utilizing independent data from a more recent survey...' – No sufficient detail on how this was done. Please expand.

L63 – ‘landscape composition and temperature in 2001 and 2016’ – unclear what the two dates refer to, is it landscape composition in 2001 and temperature in 2016, or is it both variables over both years?

L68 – ‘By subtracting the predicted species richness of the equilibrium model from the lag model, we quantified colonization credits (if the difference was positive) or extinction debts (if negative).’ Maybe I’m getting confused here but L410 actually says it the other way around? ‘Values of extinction debt and colonization credit were calculated by subtracting the predictions of species richness produced by the full (lag) model (Eq. 3) from the predicted species richness values at equilibrium (Eq. 5).’ – Please clarify.

L77-79 – can you refer to specific figures / tables / sections of the supplementary materials where these values come from?

L106 – ‘urbanization leads to the strongest lags [...] past timepoint contribution is high’ implying that community response is slow (?) – But then in L112 the authors state that ‘losses of forest caused greater lags compared to gains’ when the contribution of past timepoint is actually lower for forest loss than for gain. Am I misunderstanding or are these actually contradictory statements in terms of what a stronger/greater lag means? – And is this ‘contribution of past timepoint’ the same (or the inverse?) of what is described in L291 with regards to weights? Please clarify.

L107 – ‘10% increase in urban land leading to a present bird community still being almost completely explained by the past land cover composition’ – is it possible that bird communities are generally good at coping with urbanisation, and thus will not necessarily undergo species extinctions in the long term? Perhaps worth discussing.

Fig. 3 – conversions from forest to grassland and vice versa (in the east & west) highlight the fact that these are very dynamic landscapes. What are the implications of this, e.g. do the immigration credits / extinction debts cancel out over large geographical areas (but with community changes at smaller scales)? Perhaps worth discussing.

L148 – It’s unclear to me how both reductions and gains in wetland can be associated with colonisation credits... please explain / expand on this.

Fig. 4 – are any of the changes in land cover types strongly correlated, e.g. grassland gain and forest loss, or grassland loss and cropland gain (from looking at Fig. 3) and could this affect the model outcomes?

L197 – ‘This 15-year timeframe was selected as a reasonable scale to explore biodiversity lags to land cover change’ – a reference is needed to back up the 15-year timeframe as a reasonable timescale.

L242 – Fig. S3 only shows land cover data for 2016; it would be useful to present the same figure for 2001 too.

L251 – ‘overall observational unit length of circa 8 km’ – unclear what this means; do you mean considering the observational range from the edge of each segment?

L267 – I'd remove mention of functional groups since you didn't look at this here.

L298 – static covariates such as...?

L300 – revise wording here...

L323-326 – this is an important statement which should be made more prominent in the main text of the paper (e.g. within L165-174).

L428 – see previous comment for Fig. 4, i.e. are any of these land cover change explanatory covariates strongly correlated and could this affect the model outcomes?

L433 – no details presented on how the 2018 data was used to validate predictions of species debts / credits. Please add.

Reviewer #3 (Remarks to the Author):

This manuscript attempts to quantify unrealized gains and losses to bird species richness in response to changes in land use. The central premise—that currently observed richness is likely to lag behind the pace of land cover changes—is sound, and the modelling approach is novel. In particular, the modelling approach is quite interesting, in that it represents the equilibrium species richness as a latent variable, thus avoiding the problematic assumption in many similar studies that either past or present observations represent an equilibrium condition.

Although I am quite positive about the general approach and I think this is well-done research, I have some concerns about the presentation of the results and the framing of the manuscript. These must be addressed before a complete evaluation is possible.

1. Framing and scope of the paper

Throughout the main text, especially in the introduction, the authors frame their study in terms of bird communities. However, the authors model only changes in species richness, and so I think this framing is not particularly suitable. Throughout the main text, the authors in fact conflate community change with species richness change, which is a rather simplistic view of both communities and diversity. Community change can be rather drastic with zero richness change; as an extreme example if one land cover is replaced by another, with the bird communities also replaced (but with equal richness among communities), then the authors would frame this as zero community change, when in fact there has been 100% turnover. If the lost community is rare, then a significant conservation loss will have occurred as well; thus, from a conservation perspective, the methods in this manuscript seem likely to understate the severity of potential biodiversity change. Moreover, the term "community" at least suggests more than simple species identities; species interactions and functions are at least as important as total richness. While changes here are certainly associated with changes in richness, this is not considered in the current ms. I think some caution in framing the paper is necessary.

Secondarily, change in richness is modelled only as a function of changes in land cover, but I would hypothesize that species richness generally would be more strongly explained by land cover

variability; replacing a diverse landscape with anything homogeneous will reduce diversity. Has this been tested?

2. Presentation of results and representation of uncertainty.

Figure 1 is really nice, a very striking result. However, I am concerned about the method and presentation, especially given how much posterior uncertainty there seems to be (from looking at Figure 2). This map, as I understand it, is showing the difference between lag-model and equilibrium model species richness. But when was posterior aggregation done? I think the authors have first computed the median (or mean?) model parameters first, and then predicted in space. If so, this map could have rather large spatial biases, especially given the log link function, and it's not trivial to present posterior prediction intervals (which are needed, the signal could well be lost in the noise). The right way to do this is via posterior simulation (Gelman et al 2004, Ch 9.2). Briefly, for each posterior draw, generate a map. Then for each hexagon, compute and present the median, lower, and upper quantiles (i.e., on 3 maps). This might be infeasible for all 10,000 draws, but it should be doable on a thinned sample. Even a relatively small number of simulations will give some characterization of the uncertainty.

There is a similar problem with Figure 4. Although it's not very clearly stated (methods paragraph starting line 422), it appears that the authors are running a model with the predictions from the Bayesian model, with the parameters fixed at their mean/median. Thus, the y-variable here is not known with any kind of certainty, but rather is already modelled and comes with quite a lot of posterior uncertainty that is not included in the analysis. As with my comments for Figure 1, this analysis really needs to be repeated via posterior simulation, with the results for figure 4 updated to show credible intervals for the coefficients across all posterior runs. As presented, the standard errors for the coefficients are quite small, but this is making the assumption that the extinction-debt/colonisation-credit in each pixel is exactly equal to the mean/median prediction from the model, an assumption we know to be false. Uncertainty propagation is a must here.

3. Strength of the results

Figure 2 is the only result that properly accounts for uncertainty, and the credible intervals are quite wide. For several land use types, the intervals span nearly across the entire range from zero to one. Given the importance of the mixing parameter to all of the conclusions, I can only assume that much of this uncertainty will propagate to the other results. Thus, while the results based on the medians seem quite strong and very interesting, they may well be lost in the noise. If, when uncertainty in the results is characterized, "no observable pattern/conclusion" is included in the plausible outcomes, then it's also important to indicate the proportion of simulations had the "interesting" outcome (e.g., supported the general conclusions drawn by the paper), vs. those that showed no result or contrary patterns.

Similarly, although Figure 2 gives a decent characterization of the relative contribution of past vs present land use for various land use scenarios, I am missing a characteristic of the absolute contribution. If the mixing parameter is 1, my understanding is that 100% of *explainable* variation is explained by past land use; but if 99% of variance is unexplainable, this is still not an informative model, no? The authors mention the use of LOO for model selection—perhaps this can at least give some independent indication about whether certain parameters are informative. However, the results of the LOO procedure seem to not be presented anywhere?

Finally, credible intervals were computed from empirical quantiles with 10000 draws (line 419), but with a minimum ESS of 400 (line 397), the accuracy of the tails of these credible intervals is dubious. Although ESS is a bit of a crude measure, it still implies an average of only 10 (effective) samples for each tail. Thus I expect that simulation error/contingency can have a rather large influence on the location of these quantile intervals. Perhaps this is no problem, but the only way to know is to compute and report ESS for the mixture weight parameter. If it is quite low, then it would be advisable to either run the model for more samples or compute narrower CIs.

Minor comments

I had difficulty on the first reading to understand exactly what was meant by extinction debt/colonisation credits (with respect to the present study). I was quite confused especially by what was meant by and equilibrium model and how this was computed (e.g., line 68). Reading the methods clarified this for me, but given the structure of the ms, I think this should be clearer in the main text.

line 367 - Understandable that the spatial model is too heavy, but the authors should include an estimate of the degree to which residual spatial autocorrelation may be a problem. It would be also nice to know to what degree the route effect compensates for this, perhaps by comparing RSA in a model with and without the random effect?

Figure 2.

- I'm not sure it makes sense to show such large ranges beyond the range of the calibration data. Why not allow the x-axis to float and show a more meaningful range, e.g., calibration range +/- 25%?

- I find the explanation of the y-axis confusing, because the model structure isn't well described here or in the main text. Thus this figure doesn't really stand alone, it's necessary to dig into the methods to understand what exactly is being shown.

- After reading the paper completely and coming back to this figure, I still do not understand panel F. Is it just the coordinates of the mean value line on the plots when the x-axis is +/- 10%? If so, why include it?

- It would be nice to know more than just the range of observed changes, perhaps marginal histograms or boxplots instead of/in addition to the shaded regions?

References

Gelman, A., et al (2004). Bayesian Data Analysis (2nd edition). Chapman & Hall.

Author Rebuttal to Initial comments

Response letter

Invisible biodiversity: widespread extinction debts and colonisation credits in US bird communities

General response

We would like to thank you and the three reviewers for your comments on our manuscript and your suggestions on how to improve it. We have taken aboard most of the suggested changes, and in particular:

1. We have re-run our Bayesian analysis including all the suggested changes of reviewer 1 regarding the use of the Breeding Bird Survey (BBS) data. These include using mean rather than maximum number of species, using a new diversity metric (the effective number of species) that is less sensitive to species detectability than species richness, modelling an observer random effect, including time of day and habitat heterogeneity as additional explanatory variables. We have also run a sensitivity analysis to identify the best buffer area to calculate land use variables around each BBS transect.
2. We have rewritten some of the sections of our manuscript to deal with the comments of reviewer 2, who suggested improvements in terminology to clarify the details of the methods and the interpretation of the findings.
3. We have produced a new map to visualise the uncertainty around the predicted debts and credits presented in figure 1. This map, shown in figure S7, does not highlight any particular spatial hotspots in prediction uncertainty. Moreover, we have now propagated (as correctly suggested by reviewer 3) this uncertainty through to the GLM model examining the relationships between land cover changes and debts/credits (figure 4). Thus, the results of this model are now much more robust.

The main message of our study remains unchanged. Across the contiguous US, future extinctions and colonisations are predicted in the vast majority of the land area, implying that predictive models of future biodiversity need to take debts and credits into account if we want such predictions to be reliable. The new results also suggest a stronger impact of urbanisation on extinction debts than revealed by the previous model.

Below we provide responses to each specific comment provided by the three reviewers.

Specific responses to each comment

Reviewer #1 (Remarks to the Author):

This is a very interesting manuscript, and it has some interesting ideas. But it is a risky analysis in several ways. It is a very complicated model that works with a grossly oversimplified response variable. It plays fast and loose with many of the big data sets that are used, and misuses them in some important ways. Some of the rationales for approaches are misleading. Finally, it ends with a variety of generalizations to explain the patterns that will likely not be convincing to North American ornithologists.

RESPONSE: Thanks for your overall evaluation of our ms. Your comments really helped us to improve the use of the datasets, the main model, and thus the reliability of our results.

General comments:

1. The authors seem to not pay attention to concerns that are central to BBS analyses.
 - a. Observer differences are quite important in BBS analyses, both in estimation of population change of individual species (see Sauer et al, Kendall et al. references in this manuscript) and in species richness estimation. The text misrepresents prior research results.
 - b. Taking maximums by species over years is ad-hoc and is a flawed means of increasing detection of species, as populations are also changing between years. In general, replication over years is not a feasible strategy of accommodating issues of detectability. Evenness, as estimated from BBS counts, is not a credible metric due to varying detectability among individuals of different species.
 - c. 10-stop samples units introduce complexity into the analysis. It introduces spatial uncertainty, as no data exists on where the stops are along the route (Trust me on this; I authored the route path information used in this analysis). That error propagates along the route, if you split up the route geographic information by starting at one end and measuring along the route path.
 - d. Another complexity in the 10-stop sample unit is that BBS routes are always run sequentially from stop 1 to stop 50. That means the first 10 are early (before or at sunrise), and the last 10 are much later in the morning (3 hours after sunrise). You include a route effect, but not a time-of-day effect.
 - e. There is an extensive literature on estimation of species richness from BBS data (see papers by Nichols et al. and Boulinier et al), and there are a variety of potential multispecies occupancy approaches for analysis of BBS species richness. The authors neither cite nor consider the use these methods, even though they provide a much better alternative to ad-hoc multiyear summaries.

RESPONSE: We thank you for your comments. We have addressed the issues you have pointed to in the following ways:

- a. We have included observed ID as random effect in our model, to control for observer differences in the BBS dataset.
- b. To minimise potential detectability issues, we have: i) recalculated all the species-specific abundances using mean rather than maximum values over the two adjacent years to each of the two focal years (2010 and 2012 for 2011; 2015 and 2017 for 2016), thereby increasing the years we averaged over by one compared to the first version, from two to three years; ii) calculated the effective number of species rather than species richness, as the former is a metric that takes into account the abundance of each species, and thereby reduces the risk that easily detectable but rare species would have a disproportionate effect on our biodiversity metric; iii) excluded evenness from our model as you suggested.
- c. In regard to the spatial uncertainty of the 10-stops precise location and the error propagation along the route, we approached it by splitting the route into 5 segments of equal length and selecting for the modelling segments 1, 3 and 5. We believe this should reduce the pseudoreplication along the route and provide more independent estimates as well as reducing spatial autocorrelation (the land cover buffers do not overlap). Additionally, routes were selected as to keep only entries that followed the survey protocol accurately (RPID = 101) therefore we believe that even if some inaccuracies were to escape the methodology control the first 10 points should be approximately within the first 4.5 miles and same for the middle and final 10.
- d. We have now included time of day as a covariate in our model.

- e. We appreciate this suggestion, and we agree with you that this would be the way to go if we were to model abundance trends over time, for instance, as recently done using the same datasets (Rosenberg et al 2019 Science). However, for the purpose of our study and our model, we are really just interested in computing a biodiversity metric, not assessing species-specific trends. We have now replaced the use of species richness, which might be more sensitive to issues of detectability, and use the effective number of species, which considers species abundances and their rarity.

2. Total species strikes me as an insensitive metric when clearly different species groups respond differently to loss or gain of different habitats.

RESPONSE: We have tried to develop a new method to estimate debts and credits over a large spatial scale, and this should be the focus of our paper. Digging into the mechanisms behind the observed patterns, for instance looking into specific functional/phylogenetic responses as you suggested, would be very interesting. However, it would be an overkill for this paper, which we believe is already very long, complex and rich as it is. We have added a couple of sentences in the discussion where we expand on this aspect to emphasize that changes in the effective number of species provide only a coarse measure of biodiversity change.

3. Land use and land cover data from USGS have historically not been comparable between major releases due to changes in interpretations methods. USGS had addressed this issue by directly computing change based on reanalyses of prior releases. Is that a concern here? At a minimum, a citation is needed on comparability of categorizations over time.

RESPONSE: We are aware that a change product is also provided by the USGS databases. However, this fails to match our data needs as it considers changes based on a hierarchical system to better communicate thematic impacts. This results in the increases and decreases of land cover reported not being the actual difference in land cover between timepoints. Moreover, this product change map does not report the directionality of land cover change, which is critical for the purpose of our study. To test the validity of our comparisons between the two single land cover maps (2001 vs 2016), we use urban land cover data, the only one for which a direct comparison between single land cover map change and change product map is doable, because it is the only land cover that has only increased over time, thus avoiding the issue of directionality of change. Figure S8 shows that our comparison matches the change product map result.

4. Scale of summary is also an issue. The circular "buffers" have several disadvantages, as points run from centre to edge, and their only virtue is that all points will be inside them. But they overlap. Perhaps that is not a great concern for explanatory variables, but it merits discussion.

RESPONSE: We appreciated this comment and we think you are right. We have revisited the buffer procedure by computing different types and sizes of buffers and assessed their power in explaining the effective number of species. As you have intuitively pointed out, the best buffer appeared to be a 500m buffer around the segment (L290-296) which almost coincides with the detection range of 400m around each point count. See the new figure S2 for a clear representation of the new procedure (note that the same response applies to a similar comment down below).

Specific comments:

I. 83. What is this "novel bird data?"

RESPONSE: We have rephrased this.

I. 188. This statement is incorrect. Canada data are comparable to US data over southern Canada, so it is absolutely incorrect to say that Canada routes "are currently still being set up."

RESPONSE: We have rephrased this. What we meant is that the contiguous US data collection has been running for longer time.

I. 190. Omit "along non-linear transects." The routes follow roads, pure and simple.

RESPONSE: Done.

I. 194. Surveys are conducted Late April-July, not "May and July." June, of course, is the primary month of sampling.

RESPONSE: Corrected.

I. 197-199. This seems backward. The 15 year timeframe is the longest possible timeframe with and cover data and also are a reasonable scale for exploring biodiversity. It is not as though you chose a 15 year interval and just happened to have data in that time frame.

RESPONSE: We have rephrased this.

I. 201. What is "a sampling point?" A route? Or a stop?

RESPONSE: What we meant with sampling point was one of 5 route segments, the unit of our analysis. We have now rephrased this entire paragraph and it should be clearer.

I. 202. I am always suspicious of ad-hoc approaches that use maximum counts over years "to minimize noise in bird community data." Many factors influence BBS counts, and many analyses control for observer differences. Also, population change, sometimes dramatically, from year to year. Simply taking Max values over multiple years ignores (and introduces bias in estimation) associated with these factors.

RESPONSE: We have now reverted to using the mean, as a more conservative measure, rather than the maximum over the years.

I. 209-210. Two issues here: (1) Are these 10-stop totals the quantities that were Max'ed over the 2 years (i.e., the units called "sampling points, above")? (2) How do you find the areas associated with these 10 stops? Stops locations are not documented for the BBS? Another quite important issue is that BBS routes are conducted sequentially, so visibility and other features of diurnal activity almost certainly vary among these sample units. This does not seem to be controlled for anywhere in the analysis.

RESPONSE: Yes, the 10-stop totals make up each of the five segments (for a total of 50-stops), and each segment is the unit of our analysis. The five 10-stop totals are reported in the BBS datasets as

distinct columns, so we used these values of species abundance for our analyses. We have rephrased and restructured this entire paragraph to make our procedure clearer. Moreover, to address your last point, we have now included a time of day covariate in the model to control for variation in diurnal activity and visibility along the route.

I. 215. This is incorrect. At least 2 of two cited studies on observer differences DID find observer effects (I am a co-author on them). There is clear evidence of observer differences in the BBS, and it is clear that observer quality is increasing over time. One might argue that observer differences build in increases into BBS data, but one cannot argue that observer differences are not important.

RESPONSE: Sorry about this, it was an oversight from our part. We have now rephrased this and included an observer ID random effect to account for such observer differences.

I. 218. This is the first mention of "species richness and evenness data." You need to define them in the content of your samples.

RESPONSE: Done.

I. 221. Here is yet another term for the sample unit. Is an "observational unit" a "sampling point" and a "10 stop." Define the sampling units clearly, and use a consistent term to reference them.

RESPONSE: We have now defined our sampling unit (L238) and use these terminology consistently across the manuscript.

I. 246-253. I have several concerns with this spatial summary. (1) The areas used to assess land use overlap along routes, causing a dependence. (2) the locations of the apparent sample units (10 stop groups) are uncertain as stop locations are not known). (3) Adjacent 10 stop units within a route are not independent (i.e., a pseudoreplication concern unless accommodated in the analysis) Note..I see that this is included (I. 367), but should be mentioned here as a concern. (4). 3km seems like a very large area. A "dispersal distance" seems irrelevant for most migratory birds and resident birds, for different reasons. It is also much larger than the effective counting distance along BBS routes (400m), and is also much larger than species territories.

RESPONSE: Thanks for pointing these issues out. We have now mentioned this specific concern about pseudoreplication as you suggested us to do and revisited our modelling data by including only segments 1, 3 and 5, as to avoid any overlap between neighbour land cover buffers (L238). We have also revisited the buffer procedure by computing different types and sizes of buffers and assessed their power in explaining the effective number of species. As you have intuitively pointed out, the best buffer appeared to be a 400m buffer around the segment (L290-296).

I. 260. A "bespoke statistical model?" I had to Google this. Doesn't the word "developed" make "bespoke" redundant?

RESPONSE: Removed.

I. 279-281. Why is species richness a Poisson? As the sum of a series of variables (i.e., the occurrence of the species), I would think it would be Normal.

RESPONSE: We have now changed our explanatory variable from species richness to the effective number of species at $q=1$. This new biodiversity metric is a normally distributed continuous positive measurement.

Reviewer #2 (Remarks to the Author):

REVIEW: "Invisible biodiversity: A longitudinal gradient of extinction debts and colonization credits in US bird communities"

In this paper the authors quantify extinction debts and colonisation credits on bird communities as a result of recent (15-year) land use changes in the USA. I consider that the paper is interesting and novel within the 'ecological time lags' literature mainly because 1) it encompasses more than one land cover type, and 2) it looks at the effects habitat gains and losses (thus acknowledging that landscapes are dynamic). The authors make good use of a large dataset of bird communities (with repeated measures over two time periods) over a large geographical area. The abstract clearly communicates the key findings of the work and the figures presented in the main text are appropriate and informative. The paper is generally well-written, however, I found the language in relation to 'debts vs. credits', 'strong vs. weak lags', 'high vs. low contribution of past timepoints' rather confusing, and found myself referring to the figures (and figure legends) to try to understand what the authors meant. There also seem to be some inconsistencies in the descriptions of how credits/debts were calculated (see e.g. comments for L68 and L410). I therefore think the main text would benefit from clearer definitions and some rewording to simplify the interpretation of the findings. Additionally, I think that the discussion of the findings could be expanded to make the most of the results presented (see e.g. comments for L107 & Fig. 3) and to more prominently acknowledge that focusing on changes in species richness provides only a coarse measure of biodiversity change. Below I provide some more specific comments which I hope the authors will find useful to improve the quality of their paper.

RESPONSE: We thank you for the overall positive evaluation of our study. You make very good points which we took aboard in our attempt to revise the manuscript. In particular:

1. We have simplified and standardised the terminology across the manuscript to clear up inconsistencies and aid to the interpretation of the findings
2. We have added a few sentences in the discussion to make the point that changes in number of species over time are an important, albeit likely incomplete, aspect of biodiversity. We agree that other measures, for instance functional and taxonomic/phylogenetic diversity, could be used. However, the main scope of our paper was to provide a novel methodological approach to estimate debts and credits over a large spatial scale considering the complex dynamic of landscape change. The paper is already rich and long as it is, and adding new things will simply make it too complicated and long.

SPECIFIC COMMENTS:

L39 – 'most' community responses to habitat change are not instantaneous. In some cases they can be, e.g. if the habitat change is particularly severe/abrupt it can extirpate rare species with very small/localised populations.

RESPONSE: We have rephrased this passage accordingly.

L52-53 and also L81-89 – ‘validated our predictions utilizing independent data from a more recent survey’ [...] ‘we further validated our predictions utilizing independent data from a more recent survey...’ – No sufficient detail on how this was done. Please expand.

RESPONSE: We have expanded according to your comment.

L63 – ‘landscape composition and temperature in 2001 and 2016’ – unclear what the two dates refer to, is it landscape composition in 2001 and temperature in 2016, or is it both variables over both years?

RESPONSE: We used the landscape composition in 2001 and 2016 to estimate the contribute of the past landscape on the current number of species. Temperature in 2016 was modelled as covariate, in the model additive static unlagged covariates terms, to recognise that the number of species is affected by climatic variables, too.

L68 – ‘By subtracting the predicted species richness of the equilibrium model from the lag model, we quantified colonization credits (if the difference was positive) or extinction debts (if negative).’ Maybe I’m getting confused here but L410 actually says it the other way around? ‘Values of extinction debt and colonization credit were calculated by subtracting the predictions of species richness produced by the full (lag) model (Eq. 3) from the predicted species richness values at equilibrium (Eq. 5).’ – Please clarify.

RESPONSE: Thanks for spotting this mistake. The first sentence was wrong, it should have read “By subtracting the predicted effective number of species of the lag model from that of the equilibrium model (without the information on the past landscape), we quantified colonisation credits (if the difference was positive) or extinction debts (if negative).” We have corrected this.

L77-79 – can you refer to specific figures / tables / sections of the supplementary materials where these values come from?

RESPONSE: We have included a paragraph in the methods where we explain this procedure (L459-465).

L106 – ‘urbanization leads to the strongest lags [...] past timepoint contribution is high’ implying that community response is slow (?) – But then in L112 the authors state that ‘losses of forest caused greater lags compared to gains’ when the contribution of past timepoint is actually lower for forest loss than for gain. Am I misunderstanding or are these actually contradictory statements in terms of what a stronger/greater lag means? – And is this ‘contribution of past timepoint’ the same (or the inverse?) of what is described in L291 with regards to weights? Please clarify.

RESPONSE: We think some of these misunderstandings derived from our own inappropriate use of the word “lags”. Our model does not really estimate lags per se (or at least not as they are intended in the ecology literature). We really can only talk about the contribution of the past landscape on the current number of species. We have revised the ms bearing this difference in mind. We hope it will read more clearly now.

L107 – ‘10% increase in urban land leading to a present bird community still being almost completely explained by the past land cover composition’ – is it possible that bird communities are generally

good at coping with urbanisation, and thus will not necessarily undergo species extinctions in the long term? Perhaps worth discussing.

RESPONSE: I think you are definitely make a good point here, and definitely it would be worth discussing. But space is limited and we already had to add several new sentences to deal with yours and the other referees' comments. Since the general pattern shown in the literature is for a decrease in bird diversity due to urbanisation (see for instance Daniel Sol et al Ecology Letters, and Aronson work in Proc B), we think it is ok not to act on your suggestion in this specific case.

Fig. 3 – conversions from forest to grassland and vice versa (in the east & west) highlight the fact that these are very dynamic landscapes. What are the implications of this, e.g. do the immigration credits / extinction debts cancel out over large geographical areas (but with community changes at smaller scales)? Perhaps worth discussing.

RESPONSE: Another excellent point. What you are point out is really intriguing and one aspect that we discussed at length. However, due to space limitations and suggestions from reviewer 1 to tone down a bit the speculations about what might explain our results, or what our results might imply, we decided to refrain from adding even more speculations. We do however use the conclusions section to highlight the highly dynamic nature of landscapes and bird communities.

L148 – It's unclear to me how both reductions and gains in wetland can be associated with colonisation credits... please explain / expand on this.

RESPONSE: Due to the changes the ms has gone through, mainly a change in the response variable after the comments of referee 1, this result was not found any more and thus this comment no longer applies.

Fig. 4 – are any of the changes in land cover types strongly correlated, e.g. grassland gain and forest loss, or grassland loss and cropland gain (from looking at Fig. 3) and could this affect the model outcomes?

RESPONSE: This is definitely a good comment and we did indeed check the correlation of the land cover variables. In some instances these are high, for instance negative changes in forests are positively correlated with increases in grasslands ($t = 1448.6$, $df = 92025$, $p < 0.001$, $R=0.97$), while losses of grasslands are positively correlated with croplands increases ($t = 304.5$, $df = 92025$, $p < 0.001$, $R=0.70$). However, the inclusion or exclusion of collinear variables did not have a noticeable effect neither on the model fit nor on the spatial distribution and magnitude of debts and credits.

L197 – 'This 15-year timeframe was selected as a reasonable scale to explore biodiversity lags to land cover change' – a reference is needed to back up the 15-year timeframe as a reasonable timescale.

RESPONSE: We have rephrased this passage to make it clear that the main criterium for the selection of this time interval was data availability.

L242 – Fig. S3 only shows land cover data for 2016; it would be useful to present the same figure for 2001 too.

RESPONSE: Included.

L251 – ‘overall observational unit length of circa 8 km’ – unclear what this means; do you mean considering the observational range from the edge of each segment?

RESPONSE: We have rephrased this paragraph also in light of the comments from referee 1, and now this comment no longer applies.

L267 – I’d remove mention of functional groups since you didn’t look at this here.

RESPONSE: Done.

L298 – static covariates such as...?

RESPONSE: We have an entire section of static unlagged covariates just below this part, so we think this will be clear to the readers.

L300 – revise wording here...

RESPONSE: Done.

L323-326 – this is an important statement which should be made more prominent in the main text of the paper (e.g. within L165-174).

RESPONSE: We have added within the main text as well (L192-196).

L428 – see previous comment for Fig. 4, i.e. are any of these land cover change explanatory covariates strongly correlated and could this affect the model outcomes?

RESPONSE: See reply to the similar comment above.

L433 – no details presented on how the 2018 data was used to validate predictions of species debts / credits. Please add.

RESPONSE: This explanation has now been added in the methods.

Reviewer #3 (Remarks to the Author):

This manuscript attempts to quantify unrealized gains and losses to bird species richness in response to changes in land use. The central premise—that currently observed richness is likely to lag behind the pace of land cover changes—is sound, and the modelling approach is novel. In particular, the modelling approach is quite interesting, in that it represents the equilibrium species richness as a latent variable, thus avoiding the problematic assumption in many similar studies that either past or present observations represent an equilibrium condition.

Although I am quite positive about the general approach and I think this is well-done research, I have some concerns about the presentation of the results and the framing of the manuscript. These must be addressed before a complete evaluation is possible.

RESPONSE: Thank you for your general positive evaluation of our study, and for your very helpful comments. We have tried to address them all in the new version of our manuscript, and we provide specific answers to each of them below.

1. Framing and scope of the paper

Throughout the main text, especially in the introduction, the authors frame their study in terms of bird communities. However, the authors model only changes in species richness, and so I think this framing is not particularly suitable. Throughout the main text, the authors in fact conflate community change with species richness change, which is a rather simplistic view of both communities and diversity. Community change can be rather drastic with zero richness change; as an extreme example if one land cover is replaced by another, with the bird communities also replaced (but with equal richness among communities), then the authors would frame this as zero community change, when in fact there has been 100% turnover. If the lost community is rare, then a significant conservation loss will have occurred as well; thus, from a conservation perspective, the methods in this manuscript seem likely to understate the severity of potential biodiversity change. Moreover, the term "community" at least suggests more than simple species identities; species interactions and functions are at least as important as total richness. While changes here are certainly associated with changes in richness, this is not considered in the current ms. I think some caution in framing the paper is necessary.

RESPONSE: This is a very good point and one that we have thoroughly considered to address in the first submitted version of the paper. However, we eventually decided to only focus on number of species (now changed to EFFECTIVE number of species due to the comments of referee 1) because our main goal was to present a novel method to estimate debts and credits over large spatial scales and accounting for the dynamic nature of land cover change. This is as first necessary step to evaluate how many species will be lost/gained, but are currently unaccounted for. The next step would indeed be to look into which species are more likely to be lost, why that is, and what will be the conservation implications of that loss. But it would be too much to include all this in a single study. Here we present a novel method and provide some preliminary, interesting results. The research community can now apply and extend this approach to other important questions. We have mentioned these pitfalls and future directions in the conclusions section.

Secondarily, change in richness is modelled only as a function of changes in land cover, but I would hypothesize that species richness generally would be more strongly explained by land cover variability; replacing a diverse landscape with anything homogeneous will reduce diversity. Has this been tested?

RESPONSE: Another excellent point. While a full assessment of how changes in landscape heterogeneity affect debts and credits would be beyond the scope of the current paper, we have partly accounted for this process by including landscape heterogeneity as a covariate in our new model.

2. Presentation of results and representation of uncertainty.

Figure 1 is really nice, a very striking result. However, I am concerned about the method and presentation, especially given how much posterior uncertainty there seems to be (from looking at Figure 2). This map, as I understand it, is showing the difference between lag-model and equilibrium model species richness. But when was posterior aggregation done? I think the authors have first computed the median (or mean?) model parameters first, and then predicted in space. If so, this

map could have rather large spatial biases, especially given the log link function, and it's not trivial to present posterior prediction intervals (which are needed, the signal could well be lost in the noise). The right way to do this is via posterior simulation (Gelman et al 2004, Ch 9.2). Briefly, for each posterior draw, generate a map. Then for each hexagon, compute and present the median, lower, and upper quantiles (i.e., on 3 maps). This might be infeasible for all 10,000 draws, but it should be doable on a thinned sample. Even a relatively small number of simulations will give some characterization of the uncertainty.

RESPONSE: This is a very good point, thanks for the comment. We have now done what you have suggested, and have provided a figure showing the uncertainty for each hexagon (after 1000 draws) across the entire contiguous US by plotting the coefficient of variation (Figure S6). We think that this figure is reassuring because it does not show a strong spatial bias in uncertainty. There are large areas of very low uncertainty in the west, but these overlap with areas of very low/null debts and credits shows in figure 1, which makes sense. Everywhere else the uncertainty is still low but not spatially segregated.

There is a similar problem with Figure 4. Although it's not very clearly stated (methods paragraph starting line 422), it appears that the authors are running a model with the predictions from the Bayesian model, with the parameters fixed at their mean/median. Thus, the y-variable here is not known with any kind of certainty, but rather is already modelled and comes with quite a lot of posterior uncertainty that is not included in the analysis. As with my comments for Figure 1, this analysis really needs to be repeated via posterior simulation, with the results for figure 4 updated to show credible intervals for the coefficients across all posterior runs. As presented, the standard errors for the coefficients are quite small, but this is assuming that the extinction-debt/colonisation-credit in each pixel is exactly equal to the mean/median prediction from the model, an assumption we know to be false. Uncertainty propagation is a must here.

RESPONSE: Again, thanks for the excellent comment. We have done what you suggested and propagate the uncertainty in this analysis (L483-494). As a result the credible intervals of the model parameters in figure 4 are indeed larger as one would expect.

3. Strength of the results

Figure 2 is the only result that properly accounts for uncertainty, and the credible intervals are quite wide. For several land use types, the intervals span nearly across the entire range from zero to one. Given the importance of the mixing parameter to all of the conclusions, I can only assume that much of this uncertainty will propagate to the other results. Thus, while the results based on the medians seem quite strong and very interesting, they may well be lost in the noise. If, when uncertainty in the results is characterized, "no observable pattern/conclusion" is included in the plausible outcomes, then it's also important to indicate the proportion of simulations had the "interesting" outcome (e.g., supported the general conclusions drawn by the paper), vs. those that showed no result or contrary patterns.

RESPONSE: Thanks for this comment. We believe that this comment is rather similar to those expressed above as it points to uncertainty around the results showed in figure 1 (map of debts and credits) and figure 4 (relationship between land cover change and debts/credits). To address this concern we produced a map of the uncertainty in debts and credits for the whole US (figure S7).

Similarly, although Figure 2 gives a decent characterization of the relative contribution of past vs present land use for various land use scenarios, I am missing a characteristic of the absolute contribution. If the mixing parameter is 1, my understanding is that 100% of *explainable* variation is explained by past land use; but if 99% of variance is unexplainable, this is still not an informative model, no? The authors mention the use of LOO for model selection—perhaps this can at least give some independent indication about whether certain parameters are informative. However, the results of the LOO procedure seem to not be presented anywhere?

RESPONSE: We have not conducted the LOO procedure on the model parameters. We have conducted the LOO procedure on the choice of buffer type and size. For the purpose of model inference, we have relied on the shrinkage of model parameters and purposely assigned the parameters a prior $\beta(i,j) \sim N(0,1)$, with the aim of shrinking the parameter estimated towards zero, thus towards a very small effect.

Finally, credible intervals were computed from empirical quantiles with 10000 draws (line 419), but with a minimum ESS of 400 (line 397), the accuracy of the tails of these credible intervals is dubious. Although ESS is a bit of a crude measure, it still implies an average of only 10 (effective) samples for each tail. Thus I expect that simulation error/contingency can have a rather large influence on the location of these quantile intervals. Perhaps this is no problem, but the only way to know is to compute and report ESS for the mixture weight parameter. If it is quite low, then it would be advisable to either run the model for more samples or compute narrower CIs.

RESPONSE: Thank you for the comment. The ESS values are reported in the table S2.

Minor comments

I had difficulty on the first reading to understand exactly what was meant by extinction debt/colonisation credits (with respect to the present study). I was quite confused especially by what was meant by equilibrium model and how this was computed (e.g., line 68). Reading the methods clarified this for me, but given the structure of the ms, I think this should be clearer in the main text.

RESPONSE: We have tried to rephrase the main text to make our definitions clearer (L68-74).

line 367 - Understandable that the spatial model is too heavy, but the authors should include an estimate of the degree to which residual spatial autocorrelation may be a problem. It would be also nice to know to what degree the route effect compensates for this, perhaps by comparing RSA in a model with and without the random effect?

RESPONSE: Spatial autocorrelation was checked by running the Moran's I statistic test, which how the correlation between a certain variable varies depending on the distance in space between the points at which those variables were recorded. Spatial autocorrelation was not corrected for as Moran's I statistical analysis was not significant ($p < 0.001$) (L407-410).

Figure 2.

- I'm not sure it makes sense to show such large ranges beyond the range of the calibration data. Why not allow the x-axis to float and show a more meaningful range, e.g., calibration range +/- 25%?

RESPONSE: We have now done so following your suggestion.

- I find the explanation of the y-axis confusing, because the model structure isn't well described here or in the main text. Thus this figure doesn't really stand alone, it's necessary to dig into the methods to understand what exactly is being shown.

RESPONSE: We have rephrased the legend and changed the title of the y axis to try to aid the interpretation of this figure.

- After reading the paper completely and coming back to this figure, I still do not understand panel F. Is it just the coordinates of the mean value line on the plots when the x-axis is +/- 10%? If so, why include it?

RESPONSE: Yes, that is what it is. We believe this aids comparison of the effects of different land cover types and their directionality of change.

- It would be nice to know more than just the range of observed changes, perhaps marginal histograms or boxplots instead of/in addition to the shaded regions?

RESPONSE: Statistical summaries of land cover changes are included in table S1.

Decision Letter, first revision:

24th September 2021

Dear Dr Dominoni,

Your manuscript entitled "Invisible biodiversity: widespread extinction debts and colonisation credits in US bird communities" has now been seen again by Reviewer 2 and a new Reviewer 4, who we recruited to comment on the original concerns of Reviewer 3, who was unavailable. You will see that Reviewer 4 has some additional questions about the Bayesian analyses, which we would like to see addressed in another revision.

We therefore invite you to revise your manuscript taking into account all reviewer comments. Please highlight all changes in the manuscript text file.

* If you have not done so already please begin to revise your manuscript so that it conforms to our Article format instructions at <http://www.nature.com/natecolevol/info/final-submission>. Refer also to any guidelines provided in this letter.

[REDACTED]

We hope to receive your revised manuscript within four to eight weeks. If you cannot send it within this time, please let us know. We will be happy to consider your revision so long as nothing similar has

been accepted for publication at Nature Ecology & Evolution or published elsewhere.

Nature Ecology & Evolution is committed to improving transparency in authorship. As part of our efforts in this direction, we are now requesting that all authors identified as 'corresponding author' on published papers create and link their Open Researcher and Contributor Identifier (ORCID) with their account on the Manuscript Tracking System (MTS), prior to acceptance. ORCID helps the scientific community achieve unambiguous attribution of all scholarly contributions. You can create and link your ORCID from the home page of the MTS by clicking on 'Modify my Springer Nature account'. For more information please visit <http://www.springernature.com/orcid>.

[REDACTED]

Reviewers' comments:

Reviewer #2 (Remarks to the Author):

I thank the authors for their responses to my queries and suggestions and for the thorough revision of their paper. I found the revised version much clearer and easier to follow, partly because of the significant rewriting of large sections of the manuscript, and partly because of a more consistent use of terminology in relation to ecological time lags. I do have a few additional comments which I hope the authors will find useful to further improve the quality of their paper:

L111-119 - I fail to see how this grassland example of land cover change 'type' is different from the example of 'directionality' given for forest & grassland in L120-122 below. Please reword/clarify.

L180 - 'the legacy of the past landscapes on the current effective number are dependent...' effective number OF SPECIES?

L198 - 'avian communities are expected to re-equilibrate in the near future...' - I would add 'assuming no further changes in land cover'. Also, unclear what is meant here by 'in the near future' - within 15 years as assessed in your study? This is only a short timeframe / snapshot of biodiversity change and it may take much longer for biological communities to pay extinction debts / colonisation credits and reach a new equilibrium.

L251 - Time of day included as covariate in the legacy model, but not in the equilibrium model? Why?

L408 - Is $p < 0.001$ the significance value of the Moran's I, or is it the threshold used to assess significance? The former would indicate spatial autocorrelation (i.e. spatial distribution not random). Please clarify.

L493 – ‘four different buffer options’, but in L293 you mention six buffer ‘options’: a) a 4000m and 6000m circular buffer around the centroid of the polygon defined by the vertices of each segment, b) a 500m, 1000m, 2000m and 4000m buffer around the segment line. Please clarify.

L484 – ‘in order TO identify...’

Reviewer #4 (Remarks to the Author):

This study uses data from the North American Breeding Bird Survey (BBS) to calculate extinction debts and colonization credits based on 15 years of land use change in the continental United States. The paper uses a creative modeling approach and vastly extends our knowledge of ecological time lags by using multiple land cover types at a continental scale. The existence of debts and credits after only 15 years of change, despite all the noise in the data, is compelling evidence for their conclusions.

The authors have done an appropriate job addressing concerns about uncertainty in the results. I commend this effort, but I have some remaining suggestions that need to be addressed. These suggestions will help make the results more transparent while also helping readers correctly interpret the spatial data in Figure 1.

(1) The authors use appropriate posterior simulation methods to assess uncertainty in predictions of debts and credits across the U.S (Figure S7). However, the way that these results are presented still lead to some concerns that likely stem from a lack of clarity. Therefore, my first suggestion is that the authors include more information in the text on how the coefficient of variation (CoV) was calculated for Figure S7. What was the exact equation used (e.g., $sd / mean$, or another equation such as the Geometric CoV)? Based on existing literature, how are you interpreting ‘low’ versus ‘high’ uncertainty based on these values?

(2) Figure S7 nicely addresses the concern that there could be large spatial biases in the results. Yet, to the reader, uncertainty in estimates appears to be incredibly high. If the conventional $sd / mean$ equation was used to calculate CoV, then the uncertainty values on the map are unacceptably high. For example, a CoV of 10 means that the standard deviation is 10 times larger than the mean, and classifying a hexagon with that much uncertainty as a “credit” or “debit” is not appropriate. The scale in Figure S7 reaches values well above 100, which leads me to believe that some other equation must have been used, but readers are currently in the dark and unable to interpret these numbers.

(3) While I appreciate the use of CoV to represent uncertainty, I do think that maps showing upper and lower quantiles of the expected value would be incredibly useful to readers. Given that you have already written a script to propagate uncertainty via posterior simulation and calculate a per-hexagon CoV, it should be fairly straightforward to save the median and quartiles from each hexagon as well. Figure S7 could then become a 3-panel map showing CoV along with the lower and upper bounds of estimates.

In addition, I have several minor comments:

Currently, the wording “2880 bird communities”, which occurs throughout the manuscript, seems

ambiguous to readers. I can infer that 960 BBS routes x 3 sections along each route = 2880 different survey units. But referring to these sampling units as "communities" without an explicit definition in the main text is misleading. Some people might interpret a "community" as a single assemblage of birds associated with a particular habitat (e.g., wetland bird community), rather than a specific sampling unit. Please provide a definition and added clarity to this phrase (preferably in the results section instead of buried in the methods).

In several places (e.g., lines 80, 143, 147, etc.) regions of the U.S. are referred to as the "North East" or "South West". To remain consistent with previous literature, I suggest changing these to a single word, e.g., "Northeast" or "Southwest".

37: I suggest rewording the sentence starting with "Notably...". Many would argue that species responses to habitat change at a local scale are in fact instantaneous, as when birds respond to deforestation or wildfire. Here, non-instantons changes refer to gradual changes in species composition at the broader landscape scale.

63: Typo: "as a biodiversity metric"

125-126: There are typos in the title for Figure 2. Consider rewording to something like: "The contribution of past landscape to the current effective number of species depends on the type, amount, and ..."

139: Typo: Please change "associated to" to "associated with".

141: I suggest rewording this sentence to provide more clarity. Ex: "...much of the area in the central United States has experienced..."

Figure 4. I commend the authors for propagating uncertainty into these estimates.

209: Incorrect package name: "exactextractr"

532: This line references JAGS model code, but the main text says that models were run using STAN.

*****END*****

Author Rebuttal, first revision:

Response letter for:

Invisible biodiversity: widespread extinction debts and colonisation credits in US bird communities

Reviewers' comments:

Reviewer #2 (Remarks to the Author):

I thank the authors for their responses to my queries and suggestions and for the thorough revision of their paper. I found the revised version much clearer and easier to follow, partly because of the significant rewriting of large sections of the manuscript, and partly because of a more consistent use of terminology in relation to ecological time lags. I do have a few additional comments which I hope the authors will find useful to further improve the quality of their paper:

The authors thank the referee for noticing the effort that has gone into this revision.

L111-119 – I fail to see how this grassland example of land cover change 'type' is different from the example of 'directionality' given for forest & grassland in L120-122 below. Please reword/clarify.

We have reframed the paragraph to highlight the three main factors that appear to modulate legacy effects: the type, magnitude, and directionality of land cover changes (L105-121).

L180 – 'the legacy of the past landscapes on the current effective number are dependent...' effective number OF SPECIES?

Corrected.

L198 – 'avian communities are expected to re-equilibrate in the near future...' – I would add 'assuming no further changes in land cover'.

A valid point, now amended.

Also, unclear what is meant here by 'in the near future' – within 15 years as assessed in your study? This is only a short timeframe / snapshot of biodiversity change and it may take much longer for biological communities to pay extinction debts / colonisation credits and reach a new equilibrium.

This is a good point - thank you. Indeed, with our current approach we cannot estimate how long these debts/credits will take to realise. Therefore, we have rephrased this paragraph and avoided discussion of the duration of any relaxation period.

L251 – Time of day included as covariate in the legacy model, but not in the equilibrium model? Why?

The written explanation was incorrect, and we apologise for the confusion caused. Time of day was included as a linear effect in the static covariates function, $g(z_S; \alpha)$, which is included in both the legacy and the equilibrium models.

L408 – Is $p < 0.001$ the significance value of the Moran's I, or is it the threshold used to assess significance? The former would indicate spatial autocorrelation (i.e. spatial distribution not random). Please clarify.

We have corrected this. Specifically, the p-value was higher than 0.05. We have also included additional information on the spatial scales tested for Moran's I.

L493 – 'four different buffer options', but in L293 you mention six buffer 'options': a) a 4000m and 6000m circular buffer around the centroid of the polygon defined by the vertices of each segment, b) a 500m, 1000m, 2000m and 4000m buffer around the segment line. Please clarify.

The information at line 293 was from a previous iteration of the analysis presented in the first submitted version of the manuscript. The corrected buffers that we tested are described at line 446: a) a circular buffer around the centroid of the polygon defined by the vertices of each segment (4000m radius), b) a buffer around the segment line (500m, 2000m and 4000m radius). This is now also standardised between statements.

L484 – 'in order TO identify...'

Corrected.

Reviewer #4 (Remarks to the Author):

This study uses data from the North American Breeding Bird Survey (BBS) to calculate extinction debts and colonization credits based on 15 years of land use change in the continental United States. The paper uses a creative modeling approach and vastly extends our knowledge of ecological time lags by using multiple land cover types at a continental scale. The existence of debts and credits after only 15 years of change, despite all the noise in the data, is compelling evidence for their conclusions.

The authors have done an appropriate job addressing concerns about uncertainty in the results. I commend this effort, but I have some remaining suggestions that need to be addressed. These suggestions will help make the results more transparent while also helping readers correctly interpret the spatial data in Figure 1.

(1) The authors use appropriate posterior simulation methods to assess uncertainty in predictions of debts and credits across the U.S (Figure S7). However, the way that these results are presented still lead to some concerns that likely stem from a lack of clarity. Therefore, my first suggestion is that the authors include more information in the text on how the coefficient of variation (CoV) was calculated for Figure S7. What was

the exact equation used (e.g., $sd / mean$, or another equation such as the Geometric CoV)? Based on existing literature, how are you interpreting 'low' versus 'high' uncertainty based on these values?

The CoV used in the previous version was the standard CoV of $sd / mean$, and we agree that this should have been made explicit. However, after reading your comment, we have changed this to the geometric CoV. We have added this information and the equation used in the methods at line 466, as well as in the caption of figure S7.

(2) Figure S7 nicely addresses the concern that there could be large spatial biases in the results. Yet, to the reader, uncertainty in estimates appears to be incredibly high. If the conventional $sd / mean$ equation was used to calculate CoV, then the uncertainty values on the map are unacceptably high. For example, a CoV of 10 means that the standard deviation is 10 times larger than the mean, and classifying a hexagon with that much uncertainty as a "credit" or "debit" is not appropriate. The scale in Figure S7 reaches values well above 100, which leads me to believe that some other equation must have been used, but readers are currently in the dark and unable to interpret these numbers.

See the previous comment for clarification of which CoV was used. We also agree that our plot was not completely clear in its presentation of uncertainty. The presence of large CoV numbers was caused by a number of predictions resulting in debts and credits of magnitudes of the order of 10^{-7} . This arose in areas of minimal landscape change (and largely in desert areas of the West US), leading to mean values that were very close to zero. The propagation of uncertainty over these values resulted in large coefficients of variation for those areas. It also came to our attention that in the code we had rounded down certain intermediate outputs, causing small values of debt/credit to be zero, and thus also leading to very high CoV values. We have now removed this rounding. We have therefore produced a new uncertainty map, now using the geometric CoV, as well as upper and lower credible intervals maps of the same figure (see next comment), and included explanatory text in the caption of figure S7.

(3) While I appreciate the use of CoV to represent uncertainty, I do think that maps showing upper and lower quantiles of the expected value would be incredibly useful to readers. Given that you have already written a script to propagate uncertainty via posterior simulation and calculate a per-hexagon CoV, it should be fairly straightforward to save the median and quartiles from each hexagon as well. Figure S7 could then become a 3-panel map showing CoV along with the lower and upper bounds of estimates.

Thank you for this additional suggestion. We have now included panels to figure S7 to show the lower and upper credible intervals of our predictions and facilitate interpretation. The intervals were calculated as the lower (0.025) and upper (0.975)

quantiles from the thousand values predicted from posterior draws.

In addition, I have several minor comments:

Currently, the wording “2880 bird communities”, which occurs throughout the manuscript, seems ambiguous to readers. I can infer that 960 BBS routes x 3 sections along each route = 2880 different survey units. But referring to these sampling units as “communities” without an explicit definition in the main text is misleading. Some people might interpret a “community” as a single assemblage of birds associated with a particular habitat (e.g., wetland bird community), rather than a specific sampling unit. Please provide a definition and added clarity to this phrase (preferably in the results section instead of buried in the methods).

We have added a clarification on this point at lines 59-61: “We identified a community as the assemblage of birds associated with the landscape surrounding each survey unit (i.e. not a prespecified habitat type).”.

In several places (e.g., lines 80, 143, 147, etc.) regions of the U.S. are referred to as the “North East” or “South West”. To remain consistent with previous literature, I suggest changing these to a single word, e.g., “Northeast” or “Southwest”.

We have made these changes as suggested.

37: I suggest rewording the sentence starting with “Notably...”. Many would argue that species responses to habitat change at a local scale are in fact instantaneous, as when birds respond to deforestation or wildfire. Here, non-instantons changes refer to gradual changes in species composition at the broader landscape scale.

Rephrased accordingly.

63: Typo: “as a biodiversity metric”

Corrected.

125-126: There are typos in the title for Figure 2. Consider rewording to something like: “The contribution of past landscape to the current effective number of species depends on the type, amount, and ...”

Corrected.

139: Typo: Please change “associated to” to “associated with”.

Corrected.

141: I suggest rewording this sentence to provide more clarity. Ex: “...much of the area in the central United States has experienced...”

Adjusted.

Figure 4. I commend the authors for propagating uncertainty into these estimates. We thank the reviewer for this comment and do agree that this, together with your suggested upper and lower boundaries figure, was an important missing element.

209: Incorrect package name: "exactextractr"
Corrected.

532: This line references JAGS model code, but the main text says that models were run using STAN.
Corrected to clarify that models were run using STAN.

Decision Letter, second revision:

24th November 2021

Dear Dr. Dominoni,

Thank you for submitting your revised manuscript "Invisible biodiversity: widespread extinction debts and colonisation credits in US bird communities" (NATECOLEVOL-210112564B). It has now been seen again by Reviewer 4, whose comments are below. The reviewer finds that the paper has improved in revision, and therefore we'll be happy in principle to publish it in Nature Ecology & Evolution, pending minor revisions to satisfy the reviewers' final requests and to comply with our editorial and formatting guidelines.

[REDACTED]

Reviewer #4 (Remarks to the Author):

I have completed my review of "Invisible biodiversity: widespread extinction debts and colonisation credits in US bird communities." I thank the authors for their responses to my queries. I am satisfied with the actions and revisions that the authors have completed, and I feel that the manuscript is much

improved.

Specifically, I am satisfied with the way that the authors have addressed my concerns about uncertainty in the results (particularly Figure 1 and Figure S7). The manuscript now clearly presents the uncertainty metrics and describes the methods behind them.

I have one minor suggestion: In two places (line 476 and the caption for Figure S7) the equation for the geometric CoV is presented in the R language, rather than as a text equation. I suggest that the authors use the equation-building tools available (e.g., in Microsoft Word) to write this equation out so that it is readable even for people who are unfamiliar with R. For example, some readers may not know what "exp" or "sqrt" mean in the code.

Our ref: NATECOLEVOL-210112564B

6th December 2021

Dear Dr. Dominoni,

Thank you for your patience as we've prepared the guidelines for final submission of your Nature Ecology & Evolution manuscript, "Invisible biodiversity: widespread extinction debts and colonisation credits in US bird communities" (NATECOLEVOL-210112564B). Please carefully follow the step-by-step instructions provided in the attached file, and add a response in each row of the table to indicate the changes that you have made. Please also check and comment on any additional marked-up edits we have proposed within the text. Ensuring that each point is addressed will help to ensure that your revised manuscript can be swiftly handed over to our production team.

****We would like to start working on your revised paper, with all of the requested files and forms, as soon as possible (preferably within two weeks). Please get in contact with us immediately if you anticipate it taking more than two weeks to submit these revised files.****

In recognition of the time and expertise our reviewers provide to Nature Ecology & Evolution's editorial process, we would like to formally acknowledge their contribution to the external peer review of your manuscript entitled "Invisible biodiversity: widespread extinction debts and colonisation credits in US bird communities". For those reviewers who give their assent, we will be publishing their names

alongside the published article.

Nature Ecology & Evolution offers a Transparent Peer Review option for new original research manuscripts submitted after December 1st, 2019. As part of this initiative, we encourage our authors to support increased transparency into the peer review process by agreeing to have the reviewer comments, author rebuttal letters, and editorial decision letters published as a Supplementary item. When you submit your final files please clearly state in your cover letter whether or not you would like to participate in this initiative. Please note that failure to state your preference will result in delays in accepting your manuscript for publication.

Cover suggestions

As you prepare your final files we encourage you to consider whether you have any images or illustrations that may be appropriate for use on the cover of Nature Ecology & Evolution.

Nature Ecology & Evolution has now transitioned to a unified Rights Collection system which will allow our Author Services team to quickly and easily collect the rights and permissions required to publish your work. Approximately 10 days after your paper is formally accepted, you will receive an email in providing you with a link to complete the grant of rights. If your paper is eligible for Open Access, our Author Services team will also be in touch regarding any additional information that may be required to arrange payment for your article.

Please note that Nature Ecology & Evolution is a Transformative Journal (TJ). Authors may publish their research with us through the traditional subscription access route or make their paper immediately open access through payment of an article-processing charge (APC). Authors will not be required to make a final decision about access to their article until it has been accepted. Find out more about Transformative Journals

Authors may need to take specific actions to achieve compliance with funder and institutional open access mandates. For submissions from January 2021, if your research is supported by a funder that requires immediate open access (e.g. according to Plan S

principles) then you should select the gold OA route, and we will direct you to the compliant route where possible. For authors selecting the subscription publication route our standard licensing terms will need to be accepted, including our [self-archiving policies](https://www.springernature.com/gp/open-research/policies/journal-policies). Those standard licensing terms will supersede any other terms that the author or any third party may assert apply to any version of the manuscript.

[REDACTED]

[REDACTED]

Reviewer #4:

Remarks to the Author:

I have completed my review of "Invisible biodiversity: widespread extinction debts and colonisation credits in US bird communities." I thank the authors for their responses to my queries. I am satisfied with the actions and revisions that the authors have completed, and I feel that the manuscript is much improved.

Specifically, I am satisfied with the way that the authors have addressed my concerns about uncertainty in the results (particularly Figure 1 and Figure S7). The manuscript now clearly presents the uncertainty metrics and describes the methods behind them.

I have one minor suggestion: In two places (line 476 and the caption for Figure S7) the equation for the geometric CoV is presented in the R language, rather than as a text equation. I suggest that the authors use the equation-building tools available (e.g., in Microsoft Word) to write this equation out so that it is readable even for people who are unfamiliar with R. For example, some readers may not know what "exp" or "sqrt" mean in the code.

Author Rebuttal, second revision:

Response to Referee

Reviewer #4 (Remarks to the Author):

I have completed my review of "Invisible biodiversity: widespread extinction debts and colonisation credits in US bird communities." I thank the authors for their responses to my queries. I am satisfied with the actions and revisions that the authors have completed, and I feel that the manuscript is much improved.

Specifically, I am satisfied with the way that the authors have addressed my concerns about uncertainty in the results (particularly Figure 1 and Figure S7). The manuscript now clearly presents the uncertainty metrics and describes the methods behind them.

I have one minor suggestion: In two places (line 476 and the caption for Figure S7) the equation for the geometric CoV is presented in the R language, rather than as a text equation. I suggest that the authors use the equation-building tools available (e.g., in Microsoft Word) to write this equation out so that it is readable even for people who are unfamiliar with R. For example, some readers may not know what "exp" or "sqrt" mean in the code.

We would like to thank Reviewer 4 for their comments and positive feedback on our paper. We agree on their last suggestion, and amended the CoV equation a text equation.

Final Decision Letter:

Dear Dr Dominoni,

We are pleased to inform you that your Article entitled "Widespread extinction debts and colonisation credits in United States breeding bird communities", has now been accepted for publication in Nature Ecology & Evolution.

Over the next few weeks, your paper will be copyedited to ensure that it conforms to Nature Ecology and Evolution style. Once your paper is typeset, you will receive an email with a link to choose the appropriate publishing options for your paper and our Author Services team will be in touch regarding any additional information that may be required

You will not receive your proofs until the publishing agreement has been received through our system

Due to the importance of these deadlines, we ask you please us know now whether you will be difficult

to contact over the next month. If this is the case, we ask you provide us with the contact information (email, phone and fax) of someone who will be able to check the proofs on your behalf, and who will be available to address any last-minute problems. Once your paper has been scheduled for online publication, the Nature press office will be in touch to confirm the details.

Acceptance of your manuscript is conditional on all authors' agreement with our publication policies (see www.nature.com/authors/policies/index.html). In particular your manuscript must not be published elsewhere and there must be no announcement of the work to any media outlet until the publication date (the day on which it is uploaded onto our web site).

Please note that *Nature Ecology & Evolution* is a Transformative Journal (TJ). Authors may publish their research with us through the traditional subscription access route or make their paper immediately open access through payment of an article-processing charge (APC). Authors will not be required to make a final decision about access to their article until it has been accepted. [Find out more about Transformative Journals](https://www.springernature.com/gp/open-research/transformative-journals)

Authors may need to take specific actions to achieve compliance with funder and institutional open access mandates. For submissions from January 2021, if your research is supported by a funder that requires immediate open access (e.g. according to [Plan S principles](https://www.springernature.com/gp/open-research/plan-s-compliance)) then you should select the gold OA route, and we will direct you to the compliant route where possible. For authors selecting the subscription publication route our standard licensing terms will need to be accepted, including our [self-archiving policies](https://www.springernature.com/gp/open-research/policies/journal-policies). Those standard licensing terms will supersede any other terms that the author or any third party may assert apply to any version of the manuscript.

We welcome the submission of potential cover material (including a short caption of around 40 words) related to your manuscript; suggestions should be sent to Nature Ecology & Evolution as electronic files (the image should be 300 dpi at 210 x 297 mm in either TIFF or JPEG format). Please note that such pictures should be selected more for their aesthetic appeal than for their scientific content, and

that colour images work better than black and white or grayscale images. Please do not try to design a cover with the Nature Ecology & Evolution logo etc., and please do not submit composites of images related to your work. I am sure you will understand that we cannot make any promise as to whether any of your suggestions might be selected for the cover of the journal.

You can generate the link yourself when you receive your article DOI by entering it here: http://authors.springernature.com/share.